# Mechanisms of synthetic lethality between BRCA1/2 and 53BP1 deficiencies and DNA polymerase theta targeting

George E. Ronson [1,7], Katarzyna Starowicz [1,3,7], Elizabeth J. Anthony [1], Ann Liza Piberger[1], Lucy C. Clarke[1,4], Alexander J. Garvin [1,5], Andrew D. Beggs [1,2], Celina M. Whalley[2], Matthew J. Edmonds [1,6], James F. J. Beesley[1] & Joanna R. Morris [1] ✉

A synthetic lethal relationship exists between disruption of polymerase theta (Polθ), and loss of either 53BP1 or homologous recombination (HR) proteins, including BRCA1; however, the mechanistic basis of these observations are unclear. Here we reveal two distinct mechanisms of Polθ synthetic lethality, identifying dual influences of 1) whether Polθ is lost or inhibited, and 2) the underlying susceptible genotype. Firstly, we find that the sensitivity of BRCA1/2- and 53BP1-deficient cells to Polθ loss, and 53BP1-deficient cells to Polθ inhibition (ART558) requires RAD52, and appropriate reduction of RAD52 can ameliorate these phenotypes. We show that in the absence of Polθ, RAD52 accumulations suppress ssDNA gap-filling in G2/M and encourage MRE11 nuclease accumulation. In contrast, the survival of BRCA1-deficient cells treated with Polθ inhibitor are not restored by RAD52 suppression, and ssDNA gap-filling is prevented by the chemically inhibited polymerase itself. These data define an additional role for Polθ, reveal the mechanism underlying synthetic lethality between 53BP1, BRCA1/2 and Polθ loss, and indicate genotype-dependent Polθ inhibitor mechanisms.

Inheritance of mutations in the breast cancer predisposition *BRCA1* or *BRCA2* genes carries an elevated risk of breast and ovarian cancer. These genes are central to the process of homologous recombination (HR) repair, and current targeted therapies, such as poly(ADP-ribose) polymerase inhibitors (PARPi) and platinum-based agents, aim to exploit the HR vulnerability of BRCA-deficient tumours[1]. Cells lacking BRCA1 or BRCA2 proteins are sensitive to targeting further proteins in other repair pathways, including DNA polymerase theta (Polθ)[2,3]. Polθ, encoded by *POLQ/Polq*, also promotes the survival of cells deficient in several non-HR DNA repair pathway genes, such as *Tp53bp1*[4–6]. Polθ

consists of a large low-fidelity A-family polymerase, bearing a C-terminal polymerase domain and an N-terminal helicase domain with several roles: Polθ mediates translesion synthesis (TLS)[7] and theta-mediated end-joining (TMEJ) of DNA double-strand breaks (DSBs) in mitosis, it suppresses HR, and it has several roles in replication[2,3,8–18].

The mechanisms underlying lethality between BRCA1/2 deficiency, 53BP1 loss and Polθ targeting are emerging but currently unclear. HR and TMEJ repair mechanisms share a substrate of resected, single-stranded DNA (ssDNA) 3' ends, and may compete[2,9]. In HR-

[1]Birmingham Centre for Genome Biology and Institute of Cancer and Genomic Sciences, College of Medical and Dental Sciences, University of Birmingham, Birmingham B15 2TT, UK. [2]Genomics Birmingham, College of Medical and Dental Sciences, University of Birmingham, Birmingham B15 2TT, UK. [3]Present address: Adthera Bio, Lyndon House, 62 Hagley Road, Birmingham B16 8PE, UK. [4]Present address: West Midlands Regional Genetics Laboratory, Birmingham Women's Hospital, Mindelsohn Way, Birmingham B15 2TG, UK. [5]Present address: University of Leeds, Leeds, UK. [6]Present address: Certara Insight, Danebrook Court, Oxford Office Village, Kidlington, Oxfordshire OX5 1LQ, UK. [7]These authors contributed equally: George E. Ronson, Katarzyna Starowicz. ✉e-mail: j.morris.3@bham.ac.uk

deficient cells, Polθ targeting is associated with increased DNA breaks and ssDNA, and cell sensitivity can be reduced by suppression of DNA nucleases[16,17,19,20]. Recent reports suggest Polθ acts to fill in ssDNA gaps in nascent DNA at or immediately behind the replication fork that are associated with RAD51 and BRCA1/2-deficiencies[16–18]. The function of nucleases in promoting sensitivity to Polθ loss has been attributed to endonucleolytic cleavage at the fork[16] and to extending the ssDNA gaps in the newly made DNA[17]. 53BP1-Shieldin loss is reported both to improve lagging strand synthesis and suppress nascent DNA gaps in BRCA1-deficient cells[21], and to exacerbate their occurrence in the absence of Polθ activity[18]. In cells lacking 53BP1, and in *BRCA1/2* mutant cells, Polθ loss has been associated with increased RAD51 foci formation, leading to the notion that recombination intermediates may contribute to the death of susceptible cells[2–5].

Polθ inhibitors are able to kill *BRCA1/2*-deficient cells and cancers, including *BRCA1* mutant cells that are PARPi resistant through 53BP1-Shieldin loss[18–20,22]. Inhibition of Polθ has been suggested as an alternative, or adjunct to, PARPi or platinum-based agents to treat HR-deficient tumours and to suppress genetic revertants[19,20,23,24]. Thus, there is a need to better understand the basis of sensitivities to Polθ targeting.

Here we examined, as a core model, murine cells bearing the *Brca1*$^{C61G}$ allele and *Trp53bp1* gene loss (referred to as *53bp1* throughout)[25–27]. We find that *Brca1*$^{C61G/C61G}$ *53bp1*$^{-/-}$ cells exhibit competent HR by using non-canonical support, including through RAD52. Our data indicate that Polθ counters inappropriate RAD52-dependent activity at ssDNA gaps present late in the cell cycle, which are prominent in both BRCA1- and 53BP1-deficient cells. Consequently, RAD52 suppression restricts multiple deleterious impacts of Polθ depletion, from chromosome breaks to cell lethality. In *53bp1*$^{-/-}$ cells, sensitivity to the Polθ inhibitor, ART558, is similarly alleviated by RAD52 suppression, but intriguingly the sensitivity of *Brca1*$^{C61G/C61G}$ *53bp1*$^{-/-}$ cells to the inhibitor is not. Remarkably, in these Polθ inhibitor-treated cells, suppression of RAD52 cannot improve DNA synthesis to fill-in G2/M ssDNA gaps unless Polθ is depleted, indicating a role for the inhibited polymerase in the BRCA1-mutant context. Increasing expression of BARD1 or BRCA2 with the ability to bind RAD51 suppresses both ssDNA gaps in newly made DNA and promotes G2/M DNA synthesis fill-in in Polθ inhibitor ART558-treated BRCA1-mutant cells. These data define RAD52 as a critical determinant of synthetic lethality between BRCA1/2 or 53BP1 and Polθ loss and show different mechanisms of Polθ inhibitor sensitivity in BRCA1- versus 53BP1-deficient conditions.

## Results

### *Brca1*$^{C61G/C61G}$ *53bp1*$^{-/-}$ cells express a hypomorphic form of BRCA1

The cysteine 61 to glycine substitution within the BRCA1 RING domain causes severe but incomplete loss of protein function. The underlying human genetic mutation, c.181T>G, is classified as pathogenic[25,28]. C61G-BRCA1 cannot support the survival of haploid human cells or mouse embryos but can drive therapy resistance in otherwise BRCA1-deficient mouse tumours[26,29]. We found that crossing *Brca1*$^{+/C61G}$ mice with *53bp1*$^{-/-}$ mice resulted in *Brca1*$^{C61G/C61G}$ *53bp1*$^{-/-}$ pups born at the expected Mendelian ratios, consistent with the previously described *53bp1*$^{-/-}$ rescue of the viability of embryos homozygous for the 'RINGless' BRCA1 (*Brca1*$^{Ex2/Ex2}$, previously notated as full knock out)[30,31]. In order to confirm mutant BRCA1 protein expression, we immunoprecipitated proteins from *Brca1*$^{+/+}$ *53bp1*$^{-/-}$ and *Brca1*$^{C61G/C61G}$ *53bp1*$^{-/-}$ mouse embryonic fibroblasts (MEFs) and performed mass spectrometry analysis. Peptides identified included residues encoded by codon 61 (Supplementary Fig. 1a), verifying expression of the mutant protein. Expression levels of C61G-BRCA1 protein and its binding partner, BARD1, were ~20% that of cells with WT-BRCA1; foci of the mutant BRCA1-BARD1 heterodimer in irradiated cells were fainter; and

the number of proximity-linked ligation foci of BRCA1-BARD1 proteins was ~30% of that seen in WT cells (Fig. 1a and Supplementary Fig. 1b–d). Thus, *Brca1*$^{C61G/C61G}$ *53bp1*$^{-/-}$ cells express a reduced level of BRCA1-BARD1 heterodimer.

We next explored features of HR in *Brca1*$^{C61G/C61G}$ *53bp1*$^{-/-}$ cells. These cells exhibited BRCA1-dependent RAD51 foci formation following irradiation (IR) or cisplatin treatment and BRCA1-dependent survival, indicating the presence of functional BRCA1. Furthermore, the *Brca1*$^{C61G/C61G}$ *53bp1*$^{-/-}$ cells showed no elevation in radial chromosomes after hydroxyurea (HU) treatment and were resistant to treatment with the PARPi olaparib (Fig. 1b, c, Supplementary Fig. 1e–i). Estimating HR efficiency using semi-quantitative PCR, with primers specific for the HR outcome at a defined DSB site, indicated an HR deficit in *Brca1*$^{C61G/C61G}$ *53bp1*$^{-/-}$ cells compared to *53bp1*$^{-/-}$ cells (Fig. 1d). Quantification of sequenced PCR products from the locus showed a similar relative level of HR (Supplementary Fig. 1j). Consistent with previous reports of the HR-repressive impact of 53BP1[32] this analysis also showed that *Brca1*$^{C61G/C61G}$ *53bp1*$^{-/-}$ cells and *53bp1*$^{-/-}$ cells exhibited a greater HR outcome frequency than WT cells (Supplementary Fig. 1j). Taken together these data indicate that *Brca1*$^{C61G/C61G}$ *53bp1*$^{-/-}$ cells are HR-competent, yet perform HR less efficiently than *53bp1*$^{-/-}$ cells.

### *Brca1*$^{C61G/C61G}$ *53bp1*$^{-/-}$ cells rely on RNF168 and RAD52 to support HR

Given the reduced BRCA1:BARD1 protein levels in *Brca1*$^{C61G/C61G}$ *53bp1*$^{-/-}$ cells, we considered whether mechanisms previously reported to support HR in cells with reduced canonical HR proteins might also be employed in these cells. RNF168 promotes BRCA1-BARD1 recruitment[33] and, independently, supports RAD51 loading in BRCA1-deficient cells through PALB2 interaction[34], making RNF168 required to support HR in cells haploinsufficient for *Brca1*[35]. In *Brca1*$^{C61G/C61G}$ *53bp1*$^{-/-}$ cells, RNF168 depletion reduced both BARD1 and RAD51 foci formation following IR treatment and also reduced cell survival in otherwise untreated *Brca1*$^{C61G/C61G}$ *53bp1*$^{-/-}$ and *Brca1*$^{C61G/+}$ *53bp1*$^{-/-}$ cells, but had no impact on *53bp1*$^{-/-}$ (*Brca1*$^{+/+}$) cells (Supplementary Fig. 2a, b and Fig. 1e, f). Co-depletion with the ssDNA binding competitor of RAD51, RADX[36,37], restored both RAD51 foci numbers observed after IR treatment and cell survival to near control levels (Fig. 1f, g). These data suggest that RNF168 contributes to both BRCA1-BARD1 recruitment and supports HR in the context of *Brca1*$^{C61G/C61G}$ *53bp1*$^{-/-}$ cells.

Mammalian RAD52 (RAD52 (*S. Cerevisiae*) homologue) promotes several recombination-mediated repair and replication mechanisms (reviewed in[38]) and is essential in cells lacking BRCA1/2 or PALB2[39,40]. We assessed the reliance of *Brca1*$^{C61G/C61G}$ *53bp1*$^{-/-}$ cells on RAD52, using depletion or a small molecule inhibitor of RAD52, 6-Hydroxy-*D*L-DOPA (6-OHD), which disrupts RAD52 oligomeric ring structures and suppresses its ssDNA binding[41]. Depletion of RAD52, or treatment with 6-OHD decreased the viability of *Brca1*$^{C61G/C61G}$ *53bp1*$^{-/-}$ but not *Brca1*$^{C61G/+}$ *53bp1*$^{-/-}$ or *53bp1*$^{-/-}$ cells (Fig. 1h, j). RAD52 siRNA treatment also reduced RAD51 foci numbers in irradiated, *Brca1*$^{C61G/C61G}$ *53bp1*$^{-/-}$ S-phase cells, and co-depletion with RADX both restored RAD51 foci numbers and negated the lethality of RAD52 depletion (Fig. 1i, j). Consistent with these data, we found that HR DNA repair products were reduced by RAD52, RNF168 or BRCA1 siRNA treatments (Fig. 1k). Thus, in *Brca1*$^{C61G/C61G}$ *53bp1*$^{-/-}$ cells, both efficient HR and cell viability requires the contribution of C61G-BRCA1 together with RNF168 and RAD52.

### Synthetic lethality with Polθ loss requires RAD52

We next considered Polθ. We noted that Polθ protein levels were elevated in *53bp1*$^{-/-}$ cells compared to WT cells, and further increased in *Brca1*$^{C61G/C61G}$ *53bp1*$^{-/-}$ cells (Fig. 2a and Supplementary Fig. 3a). RAD51 foci in Polθ-depleted, but otherwise untreated *53bp1*$^{-/-}$ cells, were increased over wild-type cells, elevated further in *Brca1*$^{C61G/C61G}$ *53bp1*$^{-/-}$ cells and elevated still further in these cells 4 h after IR-exposure

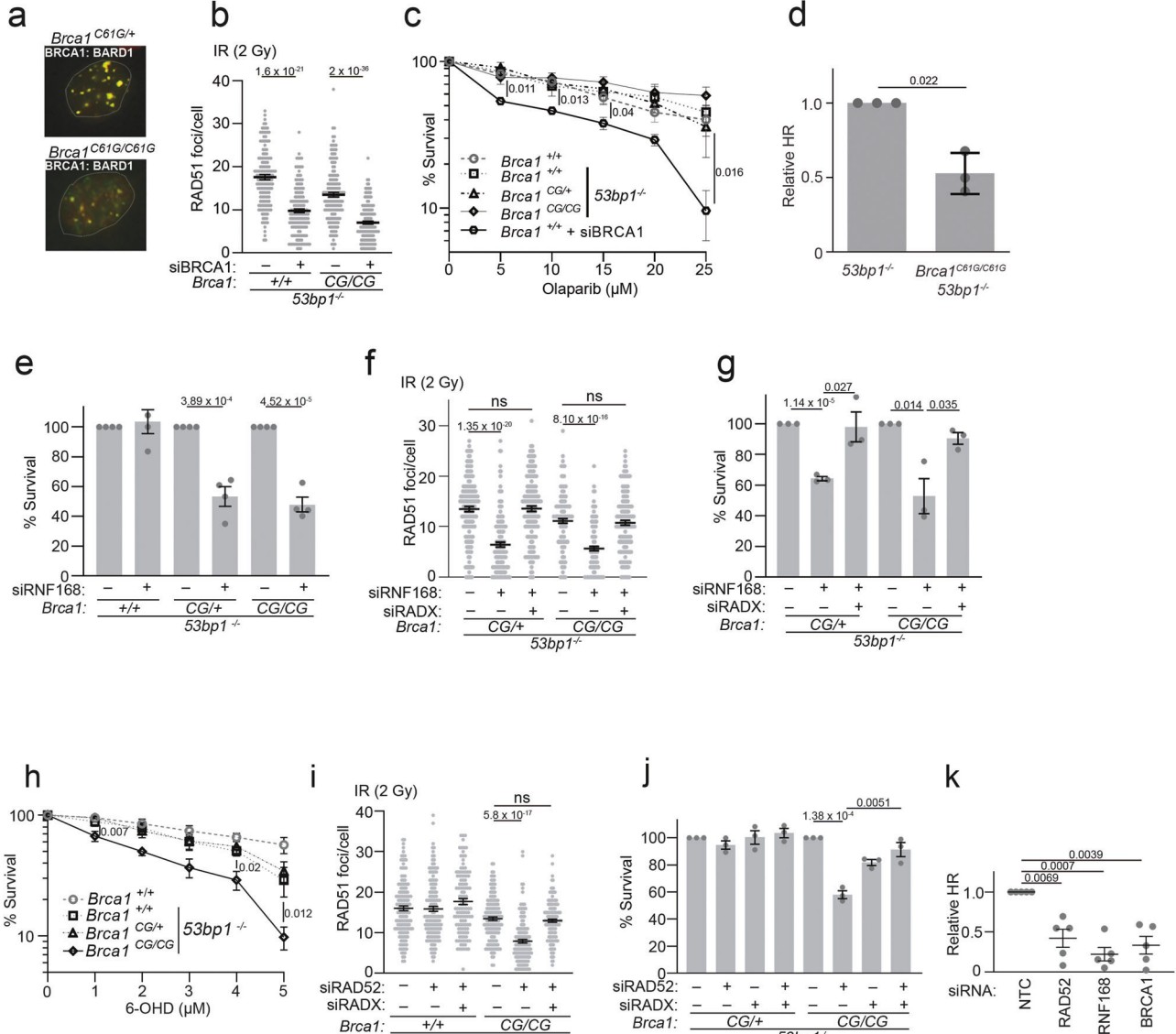

**Fig. 1 | $Brca1^{C61G/C61G}$ $53bp1^{-/-}$ cells rely on non-canonical support mechanisms for RAD51 foci, HR and survival. a** BRCA1 (red) and BARD1 (green), with yellow showing co-location, in the MEF genotypes shown (all are $53bp1^{-/-}$) fixed 3 h after exposure to 2 Gy irradiation (IR). **b** Quantification of RAD51 foci 3 hours after 2 Gy IR exposure with non-targeting control (NTC) siRNA (−) or BRCA1 siRNA (+). $n = 150$ cells from 3 biological replicates. **c** Colony survival following 16 h olaparib in MEFs of the indicated genotypes, or with BRCA1 siRNA, $n = 3$ biological replicates. **d** Quantification of HR-specific PCR product. $n = 3$ biological replicates. **e** Colony survival after treatment with NTC siRNA (−) or RNF168 siRNA (+). $n = 4$ biological repeats. **f** Quantification of RAD51 foci 3 h after 2 Gy IR exposure, of the genotypes shown (all are $53bp1^{-/-}$), following treatment with NTC siRNA (−) or siRNA to RNF168, RADX or both (+). $n = 150$ cells from 3 biological replicates, per condition.

**g** Colony survival of MEFs treated with NTC siRNA (−) or siRNA to RNF168, RADX or both (+). $n = 4$ biological repeats. **h** Colony survival of MEFs treated with the RAD52 inhibitor 6-Hydroxy-$DL$-DOPA (6-OHD). $n = 4$ biological repeats. **i** Quantification of RAD51 foci 3 h after 2 Gy IR exposure, in EdU-positive MEFs, treated with NTC siRNA (−) or siRNA to RAD52, RADX or both (+). $n = 150$ cells from 3 biological replicates, per condition. **j** Colony survival in MEFs treated with NTC siRNA (−) or siRNA to RAD52, RADX or both (+), $n = 3$ biological repeats. **k** Relative PCR product intensity of HR-specific product in $Brca1^{C61G/C61G}$ $53bp1^{-/-}$ cells treated with NTC siRNA (−) or siRNA to RAD52, RNF168 and BRCA1. $n = 5$ biological replicates. In all cases, data shown are mean ± SEM. All statistical analysis in this figure was performed using a two-tailed Student's $t$ Test, without adjustment for multiple comparisons. Source data are provided as a Source Data file.

(Fig. 2b, c). Depletion of Polθ reduced survival of $Brca1^{C61G/C61G}$ $53bp1^{-/-}$ cells to a greater degree than $53bp1^{-/-}$ cells (Supplementary Fig. 3b and Fig. 2d), and increased the appearance of γH2AX foci, a chromatin mark induced by DNA breaks, in both cell lines (Supplementary Fig. 3c). In TMEJ, the 3′ DNA flaps remaining after Polθ annealing and DNA extension are cleaved, and the strands ligated by XRCC1/ligase III[24,42]. Therefore, we assessed whether $Brca1^{C61G/C61G}$ $53bp1^{-/-}$ cells might depend more on Ligase III but noted no differential impact of the ligase I/III inhibitor, L67[43], on cell survival in otherwise untreated $Brca1^{C61G/C61G}$ $53bp1^{-/-}$ cells (Supplementary Fig. 3d).

We next investigated whether functional relationships exist between Polθ and RAD52 or RNF168 in $Brca1^{C61G/C61G}$ $53bp1^{-/-}$ cells. Polθ siRNA treatment had a negligible impact on the presence of RNF168 foci (Supplementary Fig. 4a and Fig. 3a), but increased FLAG-RAD52 accumulations in both $Brca1^{+/+}$ $53bp1^{-/-}$ and $Brca1^{C61G/C61G}$ $53bp1^{-/-}$ cells, with a significantly increased number of γH2AX foci marked by FLAG-RAD52 colocalisation in $Brca1^{C61G/C61G}$ $53bp1^{-/-}$ cells (Fig. 3b–d). In order to assess the impact of RNF168 and RAD52 on Polθ synthetic lethality, we titrated RAD52 and RNF168 siRNA onto Polθ-depleted $Brca1^{C61G/C61G}$ $53bp1^{-/-}$ cells, and examined cell survival. In addition, as an indicative

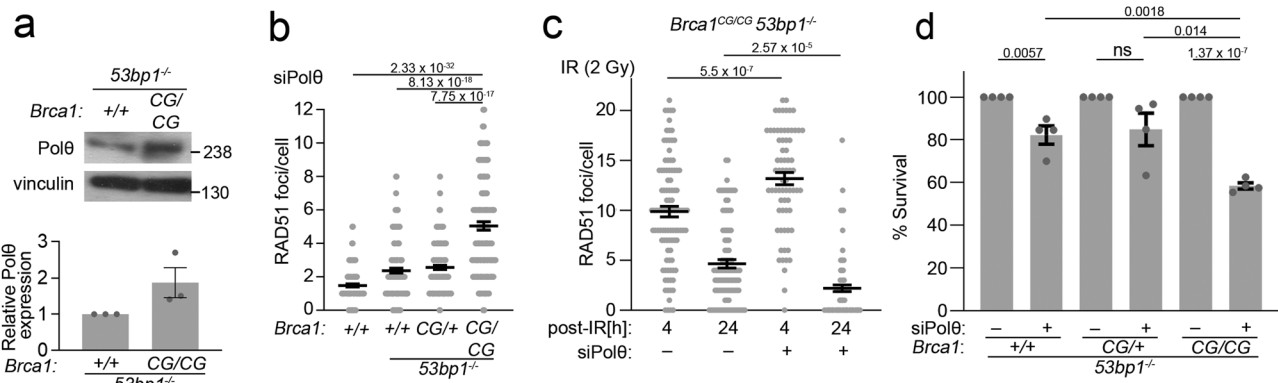

**Fig. 2 | *Brca1^{C61G/C61G} 53bp1^{-/-}* cells are sensitive to Polθ depletion. a** Western blot of Polθ and quantification of protein levels, *n* = 3 biological replicates. Data are mean ± SEM. **b** Quantification of RAD51 foci, treated with siRNA to Polθ. *n* = 100 cells from 2 biological replicates per condition. **c** Quantification of RAD51 foci 4 and 24 h after 2 Gy IR exposure, in MEFs treated with NTC siRNA (−) or siRNA targeting Polθ (+). *n* = 100 cells from 2 biological replicates, per condition. **d** Colony survival in MEFs treated with NTC siRNA (−) or siRNA targeting Polθ (+). *n* = 4 biological replicates. In all cases, data shown are mean ± SEM. All statistical analysis in this figure was performed using a two-tailed Student's *t* Test, without adjustment for multiple comparisons. Source data are provided as a Source Data file.

measure of altered resection or recombination intermediates, we examined RAD51 foci formation after IR exposure in the same siRNA-treated conditions. Low concentrations of RNF168 siRNA were able to reduce the elevated RAD51 foci numbers observed after Polθ depletion to control levels in IR-treated *Brca1^{C61G/C61G} 53bp1^{-/-}* cells, but had no impact on cell survival (Supplementary Fig. 4b, c). In contrast, low concentrations of RAD52 siRNA both suppressed RAD51 foci numbers and improved the viability of *Brca1^{C61G/C61G} 53bp1^{-/-}* cells treated with Polθ siRNA (Fig. 3e, f). Treatment with low RAD52 siRNA concentrations resulted in a partial suppression of RAD52 expression (Fig. 3g). To test these findings further, we used the RAD52 inhibitor 6-OHD. Low doses (0.3–0.035 μM) of 6-OHD also improved the survival of Polθ-depleted *Brca1^{C61G/C61G} 53bp1^{-/-}* cells and restricted the formation of RAD51 foci when these cells were irradiated (Fig. 3h, i). Similarly, the survival of cells treated with siRNA to both BRCA2 and Polθ was substantially improved by low-concentration 6-OHD treatment (Supplementary Fig. 4d, e). Extending these findings to a human system, the survival of Polθ- and BRCA1-co-depleted CAL51 human basal breast cancer epithelial cells was improved by low-concentration RAD52 inhibitor or treatment with a low concentration of RAD52 siRNA (Fig. 3j and Supplementary Fig. 4f). 6-OHD is an allosteric inhibitor of the RAD52 ssDNA binding domain that also disrupts RAD52 oligomerisation and suppresses the ability of GFP-RAD52 to form foci in cisplatin-treated cells[41]. We tested the effective dose of 6-OHD in Polθ-depleted *Brca1^{C61G/C61G} 53bp1^{-/-}* cells and noted that 0.15 μM 6-OHD reduced the average number of FLAG-RAD52 foci per cell (Fig. 3k), confirming the ability of 0.15 μM 6-OHD to partially suppress RAD52. This concentration of 6-OHD also suppressed chromosome aberrations and reduced micronuclei formation in Polθ-depleted cells (Fig. 3l, m). Intriguingly, 0.15 μM 6-OHD alone increased chromosome gaps observed in metaphase spreads, which were reduced by Polθ depletion (Fig. 3l), consistent with the recent finding that Polθ mediates some of the deleterious effects of RAD52 depletion in BRCA-deficient cells[9]. Importantly, unlike higher concentrations of 6-OHD, 0.15 μM was insufficient to increase the sensitivity of *Brca1^{C61G/C61G} 53bp1^{-/-}* cells to olaparib (Supplementary Fig. 4g). Taken together, these data suggest that RAD52 mediates many of the deleterious features associated with Polθ loss.

### RAD52 functions with RPA and MRE11 to promote the toxicity of Polθ loss

A feature of Polθ suppression in HR-deficient cells is elevated levels of ssDNA[17,18]. We reasoned that RAD52 might either contribute to the formation of ssDNA or act in a deleterious manner subsequent to ssDNA formation. The affinity of the human RPA–RAD52 complex for ssDNA is higher than that of the individual proteins in vitro and in cells, and the RPA-binding portion of RAD52 is required for its contribution to homologous recombination[44–46]. We tested concentrations of siRNA to RPA70 and RPA32 and found that 0.15–0.07 nM reduced FLAG-RAD52 foci, improved cell viability and suppressed the formation of chromosome DNA breaks and gaps in Polθ siRNA-treated *Brca1^{C61G/C61G} 53bp1^{-/-}* cells (Fig. 4a–c and Supplementary Fig. 5a). Intriguingly, as seen with low-concentration RAD52 inhibitor, these concentrations of RPA siRNA resulted in chromosomes with gaps that were suppressed by Polθ siRNA co-treatment (Fig. 4c). As RPA depletion may itself reduce resection through decreasing the activity of some nucleases[47,48], we next generated an siRNA-resistant WT FLAG-RAD52 construct and a FLAG-RAD52 variant lacking its RPA-binding region (Δ aa 254–286)[49]. We confirmed poor interaction of the FLAG-RAD52 mutant with RPA and found that it recruited poorly into foci in Polθ siRNA-treated *Brca1^{C61G/C61G} 53bp1^{-/-}* cells (Supplementary Fig. 5b–d). We tested WT-FLAG-RAD52 and Δ254−286-FLAG-RAD52 in the context of Polθ depletion and low-concentration (8 nM) RAD52 siRNA treatment, which reduces RAD52 expression to ~20% of normal levels (Fig. 3g). Importantly, the expression of WT-FLAG-RAD52 suppressed the survival advantage bestowed by RAD52 depletion, confirming that RAD52 drives the sensitivity to Polθ depletion (Fig. 4d). In contrast, expression of the Δ254−286 mutant of RAD52 did not suppress survival (Fig. 4d), suggesting that the toxicity of RAD52 in the context of Polθ depletion correlates with its ability to interact with RPA.

Depletion of DNA nucleases can improve the survival of Polθ-suppressed, BRCA1- and BRCA2-deficient cells[16,17,20]. In agreement with previous findings, we noted that low-concentration treatment with the MRE11 exonuclease inhibitor, mirin, improved the survival of *Brca1^{C61G/C61G} 53bp1^{-/-}* cells treated with Polθ siRNA (Fig. 4e). We tested the impact of mirin on RAD52 accumulations and found that concentrations sufficient to restore cell survival (2.5 μM) were insufficient to suppress RAD52 foci (Fig. 4f). These observations suggest that this degree of MRE11 suppression does not support cell survival through the inhibition of RAD52 accumulations. We considered an alternative: that RAD52 promotes MRE11 recruitment[50]. We examined Polθ-depleted *Brca1^{C61G/C61G} 53bp1^{-/-}* cells for potential proximity between MRE11 and ssDNA by growing cells in the presence of the nucleotide analogue BrdU for 48 h. BrdU is masked in dsDNA and only available to be bound by antibodies in ssDNA. Polθ siRNA treatment alone increased the detection of proximity between MRE11

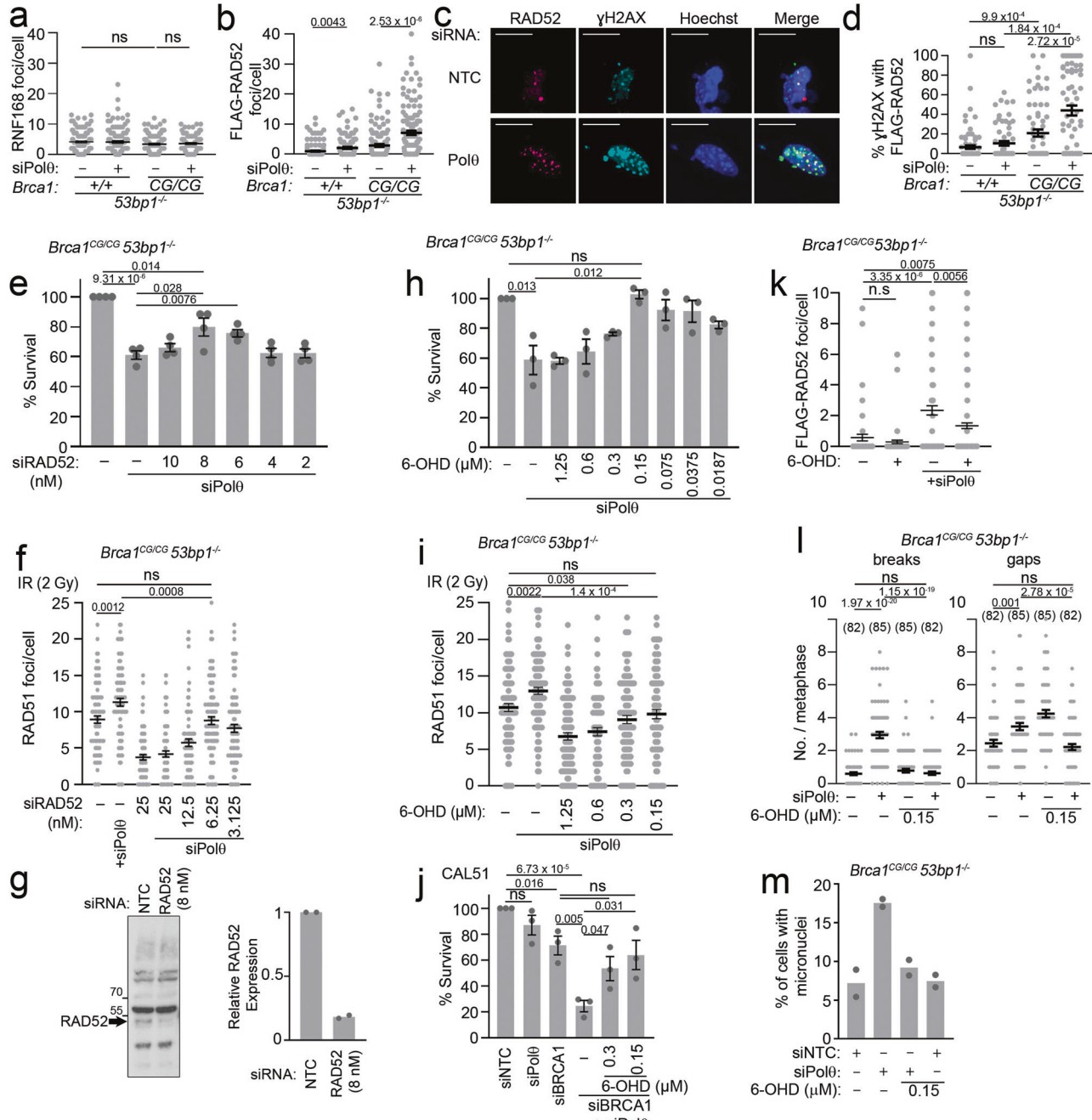

**Fig. 3 | Polθ suppresses RAD52-mediated toxicity. a** RNF168 foci, **b** FLAG-RAD52 foci in asynchronous cells with control (−) or Polθ (+) siRNA. $n = 120$ cells from 3 biological replicates. **c** RAD52 and γH2AX foci in asynchronous $Brca1^{C61G/C61G}$ $53bp1^{-/-}$ cells treated with control or Polθ siRNA. Scale bars represent 10 μm. **d** % γH2AX foci colocalised with RAD52 foci in asynchronous cells treated with control (−) or Polθ (+) siRNA. $n = 80$ cells from 2 biological replicates. **e** Colony survival of MEFs treated with Polθ, or control siRNA (−), and RAD52 siRNA. $n = 4$ biological replicates. **f** RAD51 foci in MEFs treated with IR (2 Gy) and Polθ siRNA, control (−) and RAD52 siRNA, fixed after 3 h. $n = 150$ cells from 3 biological replicates. **g** RAD52 western blot after 8 nM siRNA in $Brca1^{C61G/C61G}$ $53bp1^{-/-}$ cells. Quantification relative RAD52 in control-treated cells, $n = 2$ biological replicates. **h** Colony survival of MEFs treated with NTC or Polθ siRNA and RAD52 inhibitor 6-OHD or vehicle (−). $n = 3$ biological replicates. **i** RAD51 foci in IR (2 Gy) -treated MEFs with RAD52 inhibitor

6-OHD or vehicle (−), fixed 3 h after exposure. $n = 150$ cells from 3 biological replicates. **j** Survival of CAL51 cells treated with Polθ, BRCA1 siRNA or both, and RAD52 inhibitor 6-OHD or vehicle (−). $n = 3$ biological replicates. **k** Quantification of FLAG-RAD52 foci in cells treated with Polθ (+), or control siRNA (−) and 0.15 μM 6-OHD or vehicle. $n = 100$ cells from 3 biological replicates. **l** Number of breaks or gaps per metaphase spread, from cells treated with control (−) or Polθ (+) siRNA and RAD52 inhibitor. $n ≥ 80$ metaphases from 3 biological repeats. Data are mean ± SEM. **m** Micronuclei after treatment with Polθ, or control siRNA (siNTC) with and without RAD52 inhibitor 6-OHD. $n = 600$ cells from 2 biological replicates. In all cases, Data shown are mean ± SEM. All statistical analysis in this figure was performed using a two-tailed Student's $t$ Test, without adjustment for multiple comparisons. Source data are provided as a Source Data file.

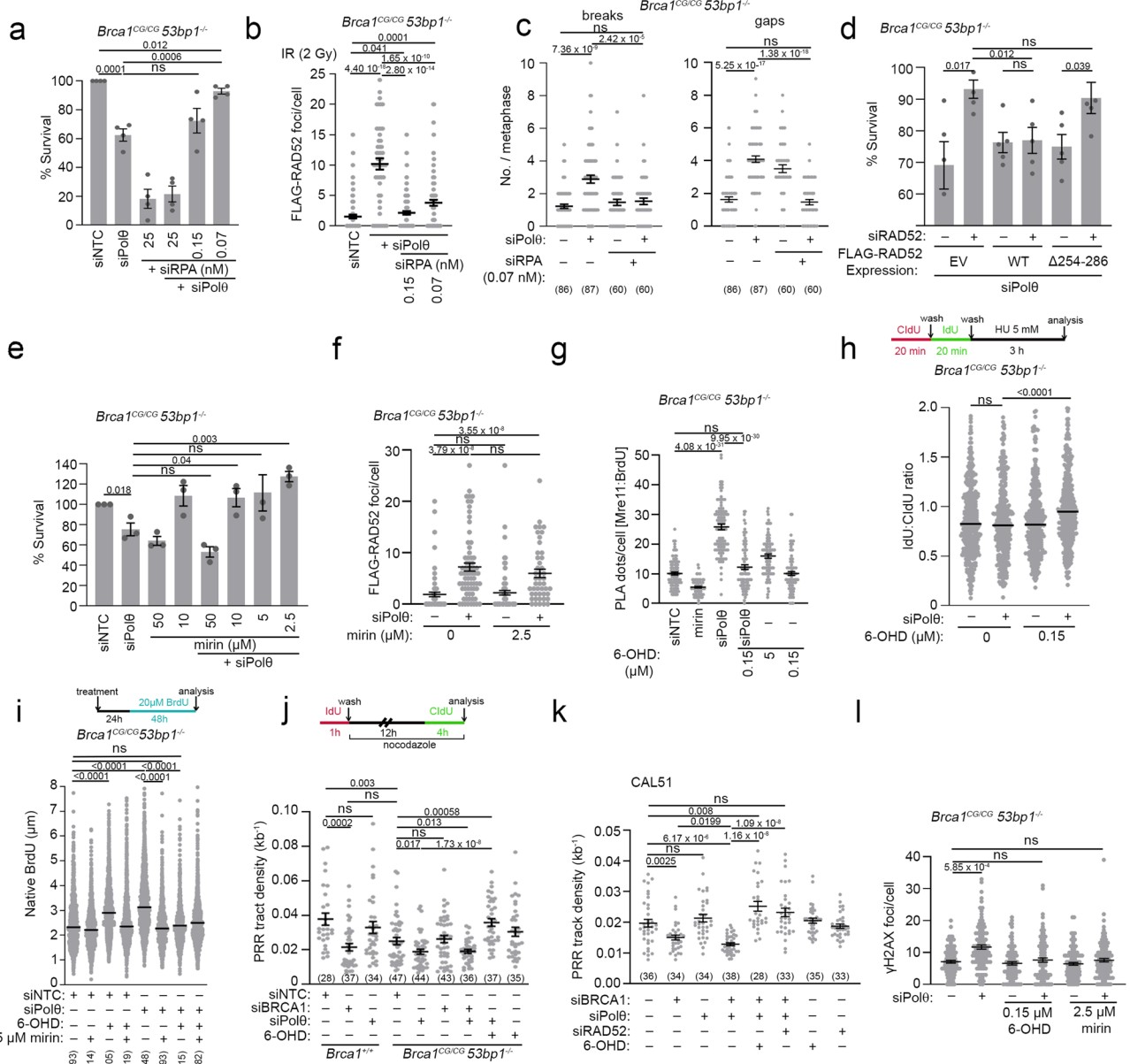

**Fig. 4 | RPA:RAD52:MRE11 suppression reduces the toxicity of Polθ depletion.** **a** Survival after control, Polθ siRNA, or Polθ siRNA with RPA siRNA. *n* = 3 biological replicates. **b** FLAG-RAD52 foci after control or Polθ siRNA with and without RPA siRNA. *n* = 100 cells from 2 biological replicates, per condition. **c** Breaks and gaps / metaphase after control (−) or Polθ (+) siRNA with or without RPA siRNA. N ≥60 metaphases from 3 biological replicates. **d** Survival of cells infected with empty (−), WT-RAD52 and aa 254−286 deleted RAD52 mutant retroviruses each resistant to RAD52 siRNA, treated with Polθ siRNA and 8 nM RAD52 (+) or control siRNA (−). *n* = 4 biological repeats. **e** Survival after Polθ siRNA and MRE11 inhibitor, mirin. n = 3 biological repeats. **f** FLAG-RAD52 foci after treatment with vehicle or 2.5 μM mirin. n = 3 biological repeats. **g** Proximity-linked ligation assay foci (PLA) between MRE11 and BrdU, treated with indicated siRNA and/or inhibitors for 72 h and with 10 μM BrdU 48 h before fixation. n = 100 cells from 2 biological replicates. **h** IdU:CldU ratios from MEFS treated with Polθ siRNA, and/or RAD52 inhibitor, 6-OHD and hydroxyurea (5 mM, 3 h). n > 350 fibres from 3 replicates. **i** Native BrdU tracts after control or Polθ siRNA and 0.15 μM RAD52 inhibitor 6-OHD and/or 5 μM mirin for 72 h. n ≥ 1400 tracks from 3 biological replicates. **j** PRR assay of MEFs with Polθ or BRCA1 siRNA and Polθ siRNA with and without RAD52 inhibitor, 6-OHD. n > 25 from 3 biological replicates. **k** PRR assay of CAL51 with Polθ or BRCA1 siRNA and Polθ siRNA with and without RAD52 inhibitor, 6-OHD. n > 25 from 3 biological replicates. **l** γH2AX foci after treatment with Polθ siRNA and RAD52 inhibitor 6-OHD or MRE11 inhibitor, mirin. n = 100 cells from 2 biological repeats. For **a**–**g** and **j**–**l**, data shown are mean ± SEM. For **h** and **l**, data shown are median. Statistical analysis in **a**–**g** and **j**–**l** was performed using a two-tailed Student's t-Test, without adjustment for multiple comparisons. Statistical analysis in **h** and **i** was performed using a Mann–Whitney test. Source data are provided as a Source Data file.

and BrdU label, whereas the combination of Polθ depletion and 0.15 μM 6-OHD RAD52 inhibitor reduced the detection of MRE11:BrdU proximity (Fig. 4g), suggesting that MRE11 is acting downstream of RAD52 when Polθ is lost in *Brca1^{C61G/C61G} 53bp1^{−/−}* cells. These data suggest that both the binding of RAD52 to RPA and the promotion of MRE11:ssDNA by RAD52 contribute to the toxicity of Polθ depletion in *Brca1^{C61G/C61G} 53bp1^{−/−}* cells.

## Polθ supports DNA synthesis late in the cell cycle
We next wished to address how Polθ functions to support cellular viability, and investigated a number of phenotypes to assess their correlation with the synthetic lethal relationship of Polθ deficiency observed in *Brca1^{C61G/C61G} 53bp1^{−/−}* cells. We considered that a phenotype of interest would be one that mirrored the relationship between RAD52 and Polθ in three important ways. Firstly, the feature must be

exacerbated by Polθ depletion; secondly, this exacerbation following Polθ loss should be rescued by the addition of RAD52 inhibitor; and finally, it should not be strongly impacted by the RAD52 inhibitor alone.

Initially, we assessed replication fork protection, replication fork restart, and the appearance of ssDNA in $Brca1^{C61G/C61G}$ $53bp1^{-/-}$ cells (Fig. 4h, i, Supplementary Fig. 5e–g). In summary, none of the outcomes of these assays correlated with the RAD52:Polθ viability relationship described above, suggesting they are unlikely to be reflective of where Polθ function is critical. For a fuller description of these results, please see the comment beneath Supplementary Fig. 5 and references[51–54].

Although elevated ssDNA lengths failed to correlate with the features of the RAD52: Polθ relationship in $Brca1^{C61G/C61G}$ $53bp1^{-/-}$ cell viability, we considered whether a subset of ssDNA regions may be relevant. As TMEJ is enriched in G2/M[9,55] and both S-phase and G2 ssDNA gaps observed in cells deficient for BRCA1/2 are MRE11-dependent[56], we hypothesised that Polθ may support DNA synthesis later in the cell cycle. We used a modified DNA fibre technique, termed the post-replication repair (PRR) assay[56]. In this assay, nascent DNA is labelled with IdU, and then nocodazole is added to arrest cells in G2/M to prevent entry into the next cell cycle. During the last 4 hours of nocodazole treatment, CldU is added to identify late DNA synthesis. We counted the number of CldU dots per tract of IdU, dividing by the total length of the tract to give the density of gap-filling PRR events per kilobase. As reported[56], BRCA1 siRNA-treated WT cells showed a reduced ability to perform fill-in DNA synthesis (Fig. 4j), and $Brca1^{C61G/C61G}$ $53bp1^{-/-}$ cells exhibited increased DNA gaps in G2/M (Supplementary Fig. 5h). $Brca1^{C61G/C61G}$ $53bp1^{-/-}$ cells also showed reduced PRR density, which was reduced further by Polθ siRNA treatment (Fig. 4j). Remarkably, fill-in DNA synthesis of Polθ siRNA-treated cells was restored by 0.15 μM 6-OHD (Fig. 4j). To confirm this observation, we examined fill-in DNA synthesis in human breast epithelial cancer cells, CAL51, and similarly observed suppression of fill-in synthesis following depletion of BRCA1. Furthermore, fill-in synthesis was further suppressed by Polθ siRNA treatment and improved by co-treatment with 0.15 μM 6-OHD RAD52 inhibitor, or by 1.5 nM RAD52 siRNA (Fig. 4k). Finally, RAD52 inhibition alone does not reduce G2/M DNA synthesis, but rather leads to a mild improvement, or no significant change, depending on the cellular model used (Fig. 4j, k). Thus, Polθ loss results in the repression of fill-in DNA synthesis in G2/M that can be restored by RAD52 suppression. These features correlate with the RAD52:Polθ relationship in cell viability. As persistent ssDNA is likely to be subject to cleavage, we examined the impact of low-concentration RAD52 or MRE11 inhibition (0.15 μM 6-OHD or 2.5 μM mirin, respectively) on the appearance of γH2AX foci, as a marker of DSB formation, in Polθ siRNA-treated $Brca1^{C61G/C61G}$ $53bp1^{-/-}$ cells. We found that both treatments suppressed γH2AX foci formation (Fig. 4l). Thus, extensive ssDNA late in the cell cycle correlates with DSBs, broken chromosomes, micronuclei formation and the toxicity of Polθ loss in $Brca1^{C61G/C61G}$ $53bp1^{-/-}$ cells.

## RAD52 is required for synthetic lethality between 53BP1 loss and Polθ inhibition

In contrast to $Brca1^{C61G/C61G}$ $53bp1^{-/-}$ cells, $53bp1^{-/-}$ cells are only mildly sensitive to Polθ siRNA (Fig. 2d). However, both cell lines show significant sensitivity to the Polθ inhibitor ART558 (Supplementary Fig. 6a). We hypothesised that the sensitivity of $53bp1^{-/-}$ cells to Polθ inhibition would be mediated by RAD52, RPA, and MRE11, as observed for the susceptibility of $Brca1^{C61G/C61G}$ $53bp1^{-/-}$ cells to Polθ loss. Indeed, co-treatment of $53bp1^{-/-}$ cells with low concentrations of RPA (RPA70/32) siRNA, low concentrations of RAD52 inhibitors, 6-OHD or D103 (which suppresses the ability of RAD52 to form foci[39]), or mirin, were each able to suppress the lethality of the Polθ inhibitor, ART558 (Supplementary Fig. 6b–e). These data confirm that ART558 toxicity in

this context requires RPA, RAD52, and MRE11. To address whether the relationship between Polθ and RAD52 can be found under alternative conditions where resection is dysregulated, we examined cells depleted for the de-ubiquitinating enzyme USP48. Cells lacking USP48 show BRCA1-dependent, extended resection in the presence of 53BP1-Shieldin[57]. USP48 siRNA-treated cells were sensitive to the Polθ inhibitor ART558, and this sensitivity was similarly ameliorated by treatment with the RAD52 inhibitor 6-OHD (Supplementary Fig. 6f, g). These findings are consistent with the idea that RAD52 contributes to the toxicity of Polθ inhibition when resection is abnormally extended.

## RAD52 is not required for the toxicity of Polθ inhibition in $Brca1^{C61G/C61G}$ $53bp1^{-/-}$ cells

$Brca1^{C61G/C61G}$ $53bp1^{-/-}$ cells are more sensitive to Polθ inhibitor than $53bp1^{-/-}$ cells (Supplementary Fig. 6a and ref. 20). Nevertheless, given that RPA:RAD52:MRE11 suppression overcame the sensitivity of $Brca1^{C61G/C61G}$ $53bp1^{-/-}$ cells to Polθ depletion and the sensitivity of $53bp1^{-/-}$ cells to Polθ inhibition, we expected the suppression of these factors to also restore the viability of Polθ inhibitor-treated $Brca1$ mutant cells. However, none of these treatments suppressed the sensitivity of $Brca1^{C61G/C61G}$ $53bp1^{-/-}$ cells to the Polθ inhibitor (Fig. 5a–d).

We considered whether the recently reported suppression of ssDNA gap filling in nascent DNA by Polθ inhibition[16–18] may reflect their increased and differential sensitivity. ssDNA gaps in nascent DNA are detectable using an adapted fibre assay utilising the S1 nuclease to cleave regions of ssDNA, resulting in a shortened nascent DNA strand[58,59]. We found that nascent DNA fibres from $Brca1^{C61G/C61G}$ $53bp1^{-/-}$ cells, but not $53bp1^{-/-}$ cells, were sensitive to the S1 nuclease (Supplementary Fig. 7a). However, we observed only a marginal increase in S1-shortened fibres in $Brca1^{C61G/C61G}$ $53bp1^{-/-}$ cells treated with Polθ inhibitor (Supplementary Fig. 7b). These data suggest that inhibited Polθ has a minimal influence on gap fill-in at these sites in this genetic background.

In order to better understand the response of $Brca1^{C61G/C61G}$ $53bp1^{-/-}$ cells to Polθ inhibition, we assessed the ability of ART558 to induce FLAG-RAD52 foci. Similar to Polθ depletion (Fig. 3b), Polθ inhibitor treatment induced RAD52 foci in both $53bp1^{-/-}$ and $Brca1^{C61G/C61G}$ $53bp1^{-/-}$ cells, with a greater induction of foci in the latter (Supplementary Fig. 7c). We then compared the ability of $53bp1^{-/-}$ and $Brca1^{C61G/C61G}$ $53bp1^{-/-}$ cells to perform G2/M DNA synthesis following ART558 treatment using the PRR assay. $53bp1^{-/-}$ cells showed a reduced ability to perform DNA synthesis compared to wild-type cells, which was further reduced following ART558 treatment and rescued by co-incubation with RAD52 inhibitor 6-OHD (Fig. 5e). In $Brca1^{C61G/C61G}$ $53bp1^{-/-}$ cells, ART558 treatment similarly suppressed fill-in DNA synthesis, but in contrast to $53bp1^{-/-}$ cells this was not rescued by co-incubation with 6-OHD (Fig. 5f). 6-OHD failed to suppress the detection of MRE11 proximity to ssDNA induced by ART558 treatment (Supplementary Fig. 7d) or the formation of micronuclei, and neither RAD52 nor MRE11 inhibitors (0.15 μM 6-OHD, 2.5 μM mirin, respectively) suppressed the generation of γH2AX foci in ART558-treated $Brca1^{C61G/C61G}$ $53bp1^{-/-}$ cells (Fig. 5g, h). In order to explain these differences, we hypothesised that the presence of the polymerase itself may be responsible for the inability of 6-OHD to rescue the viability of $Brca1^{C61G/C61G}$ $53bp1^{-/-}$ cells following ART558 treatment. Therefore, we treated $Brca1^{C61G/C61G}$ $53bp1^{-/-}$ cells depleted of Polθ with ART558. Under these conditions, the reduced fill-in DNA synthesis in Polθ-depleted and inhibitor co-treated cells was restored by 0.15 μM 6-OHD RAD52 inhibition (Fig. 5i). These data suggest that Polθ loss negates the ability of the Polθ inhibitor to prevent the restoration of G2/M gap fill-in by RAD52 inhibition. Thus, while in $53bp1^{-/-}$ cells the suppression of G2/M DNA synthesis seen when Polθ is inhibited is mediated by RAD52, the inhibited Polθ protein itself additionally suppresses DNA synthesis in $Brca1^{C61G/C61G}$ $53bp1^{-/-}$ cells. These data indicate a differential engagement of Polθ in cells without full BRCA1 function.

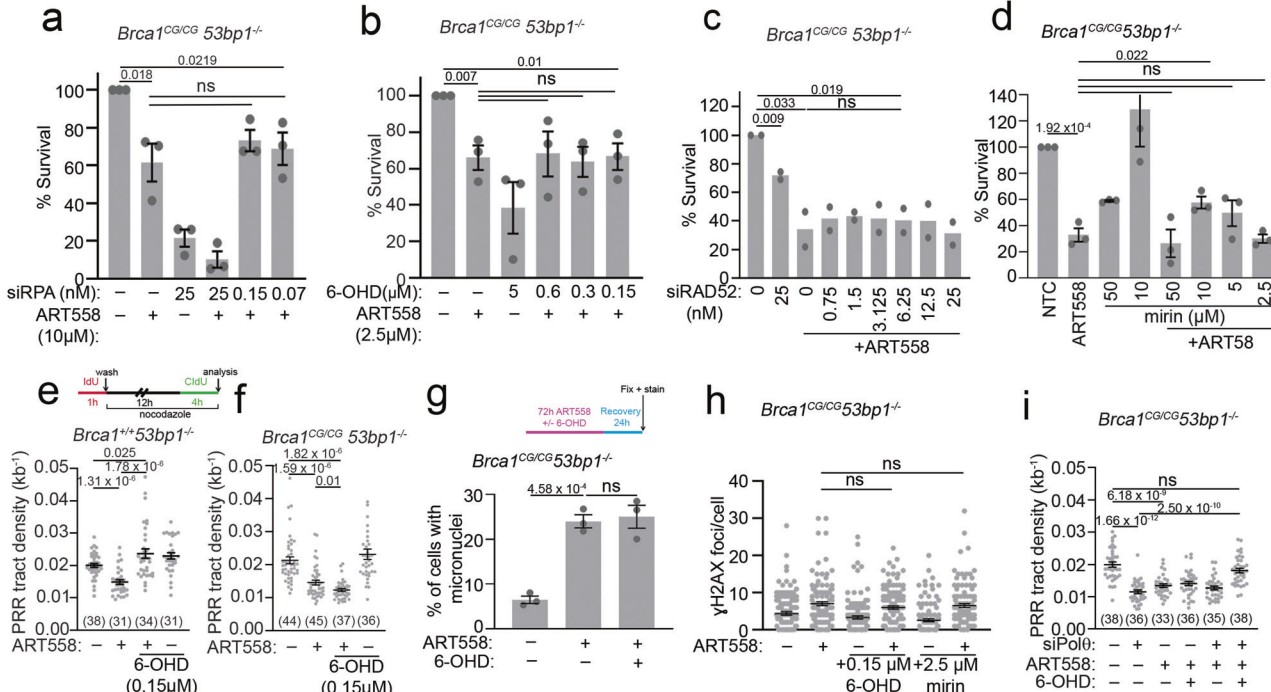

**Fig. 5 | The sensitivity of *Brca1^C61G/C61G 53bp1^-/-* cells *to* ART558 Polθ inhibition cannot be overcome by RPA:RAD52:MRE11 suppression. a** Survival following Polθ inhibitor ART558 (10 μM) and RPA siRNA. *n* = 4 biological replicates. **b** Survival following Polθ inhibitor ART558 (10 μM) and RAD52 inhibitor 6-OHD. *n* = 4 biological replicates. **c** Survival after Polθ inhibitor ART558 (10 μM) and RAD52 siRNA. *n* = 2 biological replicates. **d** Survival after 10 μM Polθ inhibitor ART558 and MRE11 inhibitor mirin. *n* = 3 biological replicates. **e** PRR density in *53bp1^-/-* cells after Polθ inhibitor ART558 (10 μM) and 0.15 μM RAD52 inhibitor 6-OHD. *n* > 30 tracks per condition from 3 biological replicates. **f** PRR density in *Brca1^C61G/C61G 53bp1^-/-* cells after Polθ inhibitor ART558 (10 μM) and 0.15 μM of the RAD52 inhibitor 6-OHD.

*N* > 30 tracks per condition from 3 biological replicates. **g** Micronuclei in cells untreated or treated with 10 μM ART558 and 0.15 μM 6-OHD or both. *n* = 600 cells per condition from 3 biological replicates. **h** γH2AX foci in cells untreated or treated with 10 μM ART558 and 0.15 μM RAD52 inhibitor 6-OHD or 2.5 μM mirin. *n* = 100 from 3 biological replicates. **i** PRR density after Polθ inhibitor ART558 (10 μM) and 0.15 μM RAD52 inhibitor 6-OHD with and without Polθ siRNA. *N* > 30 tracks per condition, from 3 biological replicates. In all cases, data shown are mean ± SEM. All statistical analysis in this figure was performed using a two-tailed Student's *t* Test, without adjustment for multiple comparisons. Source data are provided as a Source Data file.

## BRCA1-BARD1:RAD51 and BRCA2:RAD51 interactions influence Polθ synthetic lethality

To better understand the factors that drive sensitivity to inhibited Polθ, we investigated interactions between the mutant C61G-BRCA1 protein and its binding partner BARD1. To examine the interaction independently of expression levels, we first tested exogenous human C61G-BRCA1 alongside or in combination with another variant which disrupts BARD1 binding, M18T[60]. C61G and M18T substitutions each reduced the ability of BRCA1 to co-purify BARD1, or to form BARD1-induced foci, whereas the C61G-M18T-BRCA1 double mutant co-purified no detectable BARD1 and had very few BARD1-induced foci (Supplementary Fig. 7e–g). Thus human C61G-BRCA1 and BARD1 exhibit a reduced, but not entirely disrupted interaction, mirroring the reduced C61G-BRCA1:BARD1 interactions observed in *Brca1^C61G/C61G 53bp1^-/-* MEFs (Supplementary Fig. 1d). Consistent with these findings, expression of exogenous murine BARD1 increased foci numbers of the endogenous C61G-BRCA1 protein in irradiated *Brca1^C61G/C61G 53bp1^-/-* cells (from a mean of 6.5 to 12.7/cell), improved HR outcomes measured by PCR and resulted in resistance to RAD52 and Polθ depletions, the latter in a BRCA1-dependent manner (Fig. 6a–d, Supplementary Fig. 7h). Utilising this ability of exogenous BARD1 expression to rescue elements of BRCA1 dysfunction in *Brca1^C61G/C61G 53bp1^-/-* cells, we tested the expression of BARD1 mutants, including a variant that prevents BRCA1 interaction (L38R)[60] and variants suppressing nucleosome interactions (A448T, D700A)[61,62]. These mutant proteins failed to improve C61G-BRCA1 foci or promote resistance to Polθ depletion (Fig. 6e, f). In contrast, the AAE-BARD1 mutant (F125A, D127A, A128E),

which disrupts RAD51 binding[63], improved C61G-BRCA1 foci but did not improve resistance to Polθ siRNA treatment (Fig. 6e, f). These data suggest that BRCA1 recruitment through BARD1 and BARD1:RAD51 interactions contribute to protect cells from a vulnerability to Polθ depletion.

One function of BRCA1 is to promote the accumulation of PALB2-BRCA2 to sites of damage[64–66]. To explore BRCA2:RAD51 interactions, we expressed BRC4, one of the BRCA2 BRC repeats, or the C-Terminal Rad51 binding region found within exon 27 of BRCA2, each fused to the RPA subunit RPA70 (Fig. 6g). Within the context of BRCA2, BRC4 aids the exchange of RPA with RAD51[67,68]. In contrast, the BRCA2 RAD51 binding region encoded within exon 27 contacts oligomerised RAD51 to support filament stability and aid replication restart[69–71]. In agreement with previous work indicating functionality of RPA-BRC fusions[72], we found that RPA-BRC4 was able to fully restore RAD51 foci formation in irradiated S-phase cells depleted of BRCA1 (Supplementary Fig. 7i), indicating its ability to bypass BRCA1 and promote RAD51 accumulation. In contrast, RPA-Exon27 was only able to partially restore RAD51 foci formation. Strikingly, we found that the expression of RPA-BRC4, but not RPA-Exon27, overcame the toxicity of Polθ and RAD52 siRNA treatment in *Brca1^C61G/C61G 53bp1^-/-* cells (Fig. 6h). These data indicate that promoting BRCA2 BRC localisation at ssDNA independently of BRCA1 can overcome the reliance of *Brca1^C61G/C61G 53bp1^-/-* cells on Polθ and Rad52.

We next assessed the ability of either WT-BARD1 or RPA-BRC4 to suppress sensitivity of *Brca1^C61G/C61G 53bp1^-/-* cells to Polθ inhibition. Expression of either construct reduced sensitivity to ART558 and

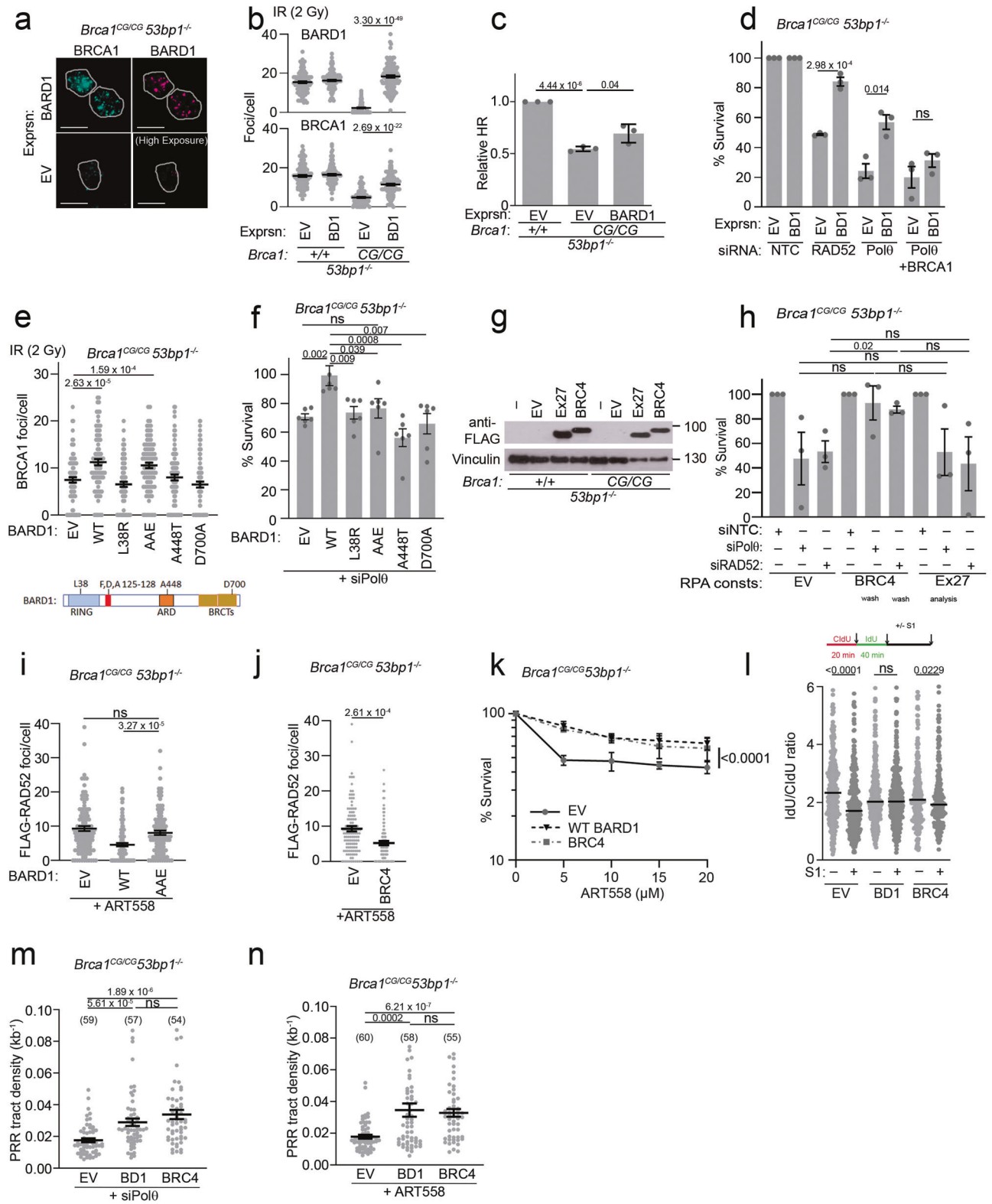

suppressed the induction of RAD52 foci following inhibitor treatment (Fig. 6i–k). Furthermore, the expression of these constructs reduced both the appearance of S1-sensitive nascent DNA in *Brca1^C61G/C61G^ 53bp1^-/-^* cells and supported the ability of these cells to perform G2/M DNA synthesis fill-in after Polθ siRNA or inhibitor treatment (Fig. 6l–n). These observations correlate the suppression of ssDNA gaps in nascent DNA and the promotion of G2/M DNA synthesis fill-in with suppression of the toxic impact of Polθ inhibition in BRCA1-mutant cells.

## Discussion

Conditions of extensive resection and poor HR repair are associated with sensitivity to Polθ loss, and further increased DNA resection has been observed around DNA DSBs in cells treated with Polθ inhibitors[16,17,19,20,73]. Polθ loss and inhibition has also been linked to increased fork-junction gaps and loss of nascent ssDNA gap fill-in[16–18]. Whether these functions are sufficient to explain the synthetic lethal relationships between HR or 53BP1 protein and Polθ loss has not been

**Fig. 6 | BRCA-RAD51 interactions suppress RAD52 recruitment and Polθ dependency. a** Endogenous C61G-BRCA1 (cyan) after 2 Gy IR with BARD1 containing (magenta) or empty vector (EV). Scale bar is 10 μm. **b** BARD1 and BRCA1 foci following infection with BARD1 containing (BD1) or empty vector (EV) after 2 Gy IR. $n = 110$ from 3 biological repeats. **c** Relative HR PCR product after BD1 or EV retroviral infection. $n = 3$ biological replicates. **d** Survival after control, RAD52, Polθ, or Polθ and BRCA1 siRNA and BD1 or EV retroviral infection. $n = 3$ biological repeats. **e** BRCA1 foci after EV or BARD1 mutant retrovirus, mutants illustrated bellow. $n = 90$, across 3 biological repeats. **f** Survival after Polθ siRNA and retrovirus infection as in **e**. $n = 6$ biological repeats. **g** Western blot of cells infected with EV or retrovirus expressing RPA-70-BRCA2-Exon27 (Ex27) or RPA-70-BRCA2-BRC4 (BRC4). **h** Survival after transfection with NTC, Polθ or RAD52 siRNA and infection with empty retrovirus or those expressing RPA constructs as indicated (RPA consts). $n = 3$ biological repeats. **i** FLAG-RAD52 foci after WT-BARD1 or AAE-BARD1

mutant retrovirus infection and 10 μM ART558. $n = 100$ from 3 biological repeats. **j** FLAG-RAD52 foci after infection with EV or BRC4 retrovirus and 10 μM ART558. $n = 90$ from 3 biological repeats. **k** Survival after EV, BRC4 or Ex27 retrovirus infection and Polθ inhibitor ART558. $n = 3$ biological repeats. **l** S1 nuclease assay of nascent DNA after EV, WT-BARD1 (BD1) or BRC4 infection. $N \geq 260$ tracks from 3 biological replicates per condition. **m** PRR density after EV, WT-BARD1 (BD1) or BRC4 retroviral infection and siPolθ siRNA. $N \geq 50$ tracks from 3 biological replicates. **n** PRR density after EV or WT-BARD1 (BD1) or BRC4 and Polθ inhibitor ART558 (10 μM). $n \geq 50$ tracks from 3 biological replicates. For **b**–**f**, **h**–**k**, **m** and **n**, data shown are mean ± SEM. For **l**, data shown are median. Statistical analysis in **b**–**f**, **h**–**j**, **m** and **n** was performed using a two-tailed Student's t Test, without adjustment for multiple comparisons. Statistical analysis in **k** was performed using a two-way ANOVA. Statistical analysis in **l** was performed using a Mann–Whitney test. Source data are provided as a Source Data file.

clear. For example, suppression of the MRE11 exonuclease can improve cell survival of Polθ-targeted cells, yet junction gaps and nascent DNA gaps are not repressed by inhibition of the exonuclease[16,17,69,74]. Further, $53bp1^{-/-}$ cells are sensitive to Polθ targeting, yet these cells do not exhibit S1-sensitive nascent DNA indicative of junction gaps or gaps in nascent DNA. Here, we define another role for Polθ. We observed that Polθ promotes fill-in DNA synthesis of ssDNA in G2/M in $53bp1^{-/-}$, BRCA1-deficient, and $Brca1^{C61G/C61G}\ 53bp1^{-/-}$ cells. Intriguingly, a recent assessment of Polθ accumulations throughout the cell cycle has highlighted that it forms foci in G2[75]. The source of the G2/M gaps is unclear. We speculate that 53BP1 may contribute to the suppression of resection at ssDNA gaps arising through excision repair pathways as it does at DNA breaks[76], and therefore also suppresses excessive RAD52 engagement. In BRCA-deficient cells, gaps arise from problems at the replication fork and may also arise through poor RAD51-mediated protection and repair of gaps that are formed through other means through the cell cycle. In $53bp1^{-/-}$ cells, the impact of the Polθ inhibitor, ART558, as well as the need for Polθ in gap-filling DNA synthesis, is overcome by RAD52 suppression, suggesting that Polθ is not itself needed for the DNA synthesis but instead counters deleterious RAD52-mediated functions. The resolution of post-replicative gaps in G2 has recently been attributed to the translesion polymerases REV1:POLζ[56,77]. As the helicase domain of Polθ strips RPA from ssDNA[8,78], we speculate that the Polθ helicase activity may suppress RPA:RAD52 accumulation. Whether ART558, an allosteric inhibitor of the polymerase domain, reduces helicase processivity is unknown, but such activity would be consistent with our findings.

RAD52 has critical functions in promoting single-strand annealing when resection is abnormally extended and in supporting residual HR and stalled replication fork restart of HR-defective cells[57,79–83]. We find that high-concentration inhibition of RAD52 (or RPA siRNA or MRE11 inhibition) is incompatible with Polθ loss, consistent with the finding that *Polq* and *Rad52* gene losses are synthetic lethal[4]. Similarly, we find that a high concentration of RAD52 inhibition or siRNA treatment is incompatible with $Brca1^{C61G/C61G}\ 53bp1^{-/-}$ cell survival, consistent with the requirement for RAD52 to support HR and single-strand annealing. Our data suggest that the loss of Polθ in BRCA1/2-, 53BP1- or USP48-deficient backgrounds exposes cells to a harmful function of RAD52: to promote increased resection and suppress G2/M ssDNA gap fill-in. Partial depletion or low-concentration inhibition of RPA:RAD52:MRE11 can overcome the need for Polθ, without exposing the cell to the vulnerabilities of RPA:RAD52:MRE11 loss, suggesting that high concentrations or activities of these proteins are deleterious in the context of Polθ depletion.

When RAD52 binds ssDNA, it does not displace RPA[49]. Since RPA enhances some nuclease activities, RAD52 engagement may encourage further nuclease interactions and activities[47,48]. Our data suggest that MRE11 is downstream of RAD52, but we do not discount the possibility that RAD52 promotes other nucleases, since a limitation of our study is that RAD52 suppression may restrict MRE11 recruitment

due to reduced resection. Indeed, the sensitivity of Polθ-inhibitor treated $Brca1^{\Delta11/\Delta11}53bp1^{-/-}$ cells can be suppressed by EXO1 reduction[20]. Similarly, we do not discount a role for recombination or RAD51, the recruitment and activity of which can be mediated by mammalian RAD52[84]. RAD52 can promote the annealing of RPA-coated ssDNA[49] and may inappropriately 'capture' nearby ssDNA bearing similar sequences, contributing to toxic intermediates. We observe that both excessive RAD51 foci formation and extended RAD51 foci kinetics are also corrected by low-concentration RAD52 inhibition in irradiated Polθ-depleted $Brca1^{C61G/C61G}\ 53bp1^{-/-}$cells. Our findings are consistent with the idea that persistent ssDNA gaps are degraded or processed into DSBs and chromosome breaks in G2 (Fig. 7). Moreover, since DSBs occurring in G2/M may be typically repaired by TMEJ[9,55], cells lacking Polθ activity would be expected to be highly sensitive to these lesions.

Our data indicate that ART558-inhibited Polθ is differently deleterious to $Brca1^{C61G/C61G}\ 53bp1^{-/-}$ versus $53bp1^{-/-}$ cells. Whereas the sensitivity of $53bp1^{-/-}$ cells to the inhibitor can be suppressed by dampening RPA:RAD52:MRE11, the sensitivity of $Brca1^{C61G/C61G}\ 53bp1^{-/-}$ cells to ART558 by these means cannot. The work of the Lord lab and colleagues recently showed that ART558 treatment increased the residence time of YFP-tagged Polθ at laser-induced DNA damage sites, consistent with possible trapping of Polθ on DNA by the compound[20]. Here we find that ART558-treated $Brca1^{C61G/C61G}\ 53bp1^{-/-}$ cells maintain MRE11:ssDNA association regardless of RAD52 suppression, and that G2/M DNA synthesis fill-in in ART558-treated cells becomes possible after RAD52 suppression once Polθ is depleted, suggesting a role for the inhibited polymerase. As RPA-BRC4 and WT-BARD1 expression suppress both the appearance of nascent ssDNA gaps and promote G2/M fill-in DNA synthesis, we cannot yet indicate whether the gaps at or behind the fork or gaps occurring later in G2/M represent the likely site of toxic Polθ engagement. Each is plausible. In the absence of adequate RAD51 at the fork, a proportion of Polθ may be engaged and trapped at junctions when the polymerase is inhibited (Fig. 7). BRC4 is implicated in the RAD51:Polα interaction and the suppression of spontaneous gaps[69], and it may be relevant that the polymerase domain of Polθ inserts and extends DNA synthesis opposite nucleotide lesions at the fork[15] and can suppress ssDNA at the fork[16]. Alternatively, or additionally, Polθ may be engaged differentially at G2/M gaps in the absence of RPA exchange and RAD51 loading via BRCA1/2 proteins.

The recently described Polθ inhibitor RP-6685, made by RePARE therapeutics, is an agent that interacts with and inhibits DNA-bound Polθ[22]. Recent papers have described the cellular responses to a Polθ inhibitor based on a recently published family of Polθ inhibitors, named 'Polθi'[16,17,85,86]. Whether 'Polθi' or RP-6685 have similar discrimination between 53BP1 and BRCA deficiencies awaits further investigation. Recent papers describing the cellular responses to 'Polθi' found that lethality with BRCA1 or BRCA2 loss was suppressed by loss of the MRE11:NBS1:RAD50 complex or by the MRE11 partner CtIP[16,17,85,86]. Some of these effects relate to endonucleolytic MRE11

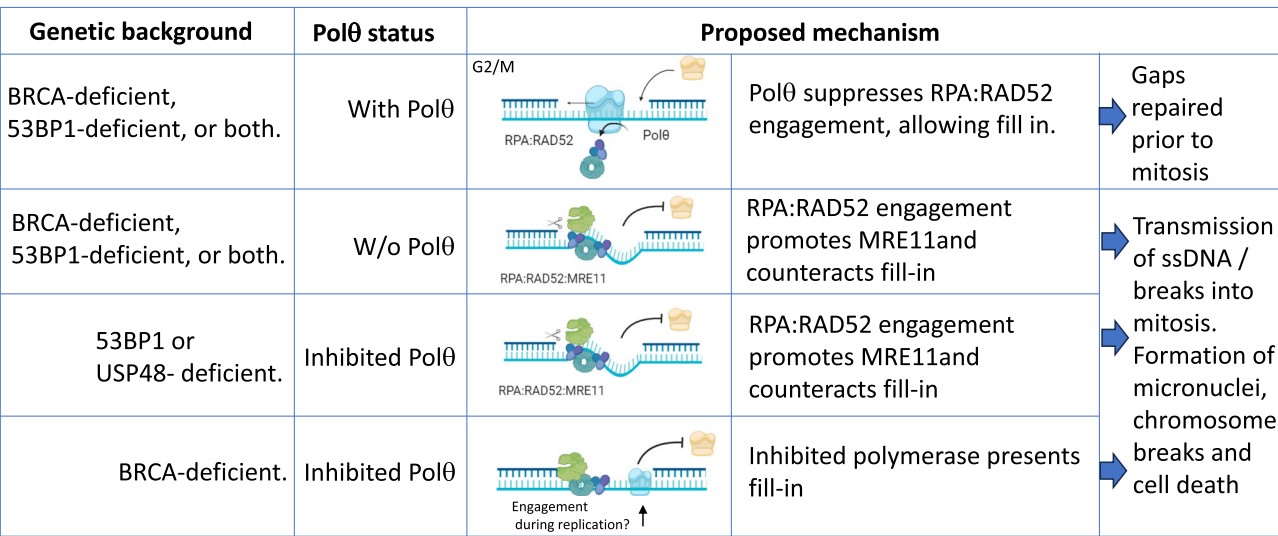

| Genetic background | Polθ status | Proposed mechanism | | |
|---|---|---|---|---|
| BRCA-deficient, 53BP1-deficient, or both. | With Polθ | | Polθ suppresses RPA:RAD52 engagement, allowing fill in. | Gaps repaired prior to mitosis |
| BRCA-deficient, 53BP1-deficient, or both. | W/o Polθ | | RPA:RAD52 engagement promotes MRE11and counteracts fill-in | Transmission of ssDNA / breaks into mitosis. Formation of micronuclei, chromosome breaks and cell death |
| 53BP1 or USP48- deficient. | Inhibited Polθ | | RPA:RAD52 engagement promotes MRE11and counteracts fill-in | |
| BRCA-deficient. | Inhibited Polθ | | Inhibited polymerase presents fill-in | |

**Fig. 7 | Proposed model of synthetic lethal mechanisms of Polθ loss and inhibition with 53BP1 and BRCA1/2 deficiencies.** Illustration of the proposed mechanism for sensitivity to Polθ loss or its inhibition according to genotype. Created with BioRender.com.

cleavage at the stalled fork junction itself[16]. We do not currently understand why, in contrast to recent reports of BRCA1-deficiency[17,18], we do not observe an increase in nascent ssDNA gaps in *Brca1^{C61G/C61G} 53bp1^{-/-}* cells treated with Polθ inhibitor. Due to this difference, we cannot discount a role of inhibited Polθ in processing these structures. We can nevertheless conclude that additional mechanisms of Polθ sensitivity exist, described herein.

In summary, we reveal a surprisingly unifying finding; that RAD52 mediates the toxicity of Polθ inhibition associated with 53BP1 or USP48 deficiencies and the toxicity of Polθ depletion in 53BP1- and BRCA1/2-deficient cells. In these contexts, RAD52 suppresses G2/M fill-in DNA synthesis and promotes DNA resection, markers of DSBs, chromosome breaks, micronuclei, and cell death. Thus, limiting RPA:RAD52:MRE11 function can overcome the gap-filling defect and cell sensitivity to Polθ suppression. In contrast, BRCA1-mutant cells retain MRE11:ssDNA interaction, poor G2/M gap fill-in, DSB markers, micronuclei and sensitivity in response to ART558, whether or not RAD52 is suppressed. We find that depletion of Polθ in these Polθ-inhibitor treated cells rescues the ability to restore DNA synthesis fill-in by RAD52 inhibition, revealing a role for the chemically inhibited Polθ in BRCA1-mutant cells. RAD51-mediated suppression of nascent DNA gaps and promotion of G2/M fill-in correlates with resistance to the Polθ inhibitor, leading us to suggest that inhibited Polθ is engaged differently when BRCA proteins are absent. These findings are likely to be important in the future development of Polθ inhibitors in considering which patients to treat with Polθ inhibitors, and have implications for the likelihood and mechanisms of resistance.

## Methods

### Animal statement
The generation of *Brca1* C61G allele is previously described[26]. Mice with the Trp53bp1[+/- 87] allele were obtained from the NIH (Bethesda). The Research Ethics Committee for animal experimentation at the University of Birmingham, UK reviewed and the Home Office approved all the work included in this manuscript. All in vivo experiments were performed under the UK Animals (Scientific Procedures) Act 1986 Home Office regulations under the authority of PPL70/8013.

### Cell line maintenance and Generation of Mouse Embryonic Fibroblasts
*Brca1^{C61G/+} S3bp1^{-/-}* male and female animals were mated to generate littermates of required *Brca1* genotypes. Pregnant mice were euthanised 13.5 days after mating and the embryos were dissected into media to allow fibroblasts to grow out. MEFs were immortalised by transduction with the SV40 large T antigen (pBsSVD2005, AdGene) using FuGENE (Promega). FlpIn U2OS and FlpIn HEK293T cell lines were from Morris lab stocks. MEFs, HEK293 and U2OS cells were maintained Dulbecco's modified Eagles medium (DMEM) supplemented with 10 % fetal bovine serum (FBS) and 1 % penicillin and streptomycin.

### Mass spectrometric analysis
Mouse BRCA1 was immunoprecipitated with the mBRCA1-N-terminal antibody (C40) and the precipitate was briefly run into a polyacrylamide Tris Acetate gradient gel (Invitrogen). The sample was excised into 10 slices, which were washed with 25 mM ammonium bicarbonate followed by acetonitrile. Following this, samples were reduced with 10 mM dithiothreitol at 60 °C followed by alkylation with 50 mM iodoacetamide at RT. Subsequently, samples were digested with trypsin (Promega) at 37 °C for 4 h. Finally, samples were quenched with formic acid and the supernatant was analysed directly without further processing.

Each gel digest was analysed by nano LC/MS/MS with a Waters NanoAcquity HPLC system interfaced to a ThermoFisher Q Exactive. Peptides were loaded on a trapping column and eluted over a 75 μm analytical column at 350 nL/min; both columns were packed with Luna C18 resin (Phenomenex). The mass spectrometer was operated in data-dependent mode, with MS and MS/MS performed in the Orbitrap at 70,000 FWHM and 17,500 FWHM resolution, respectively. The fifteen most abundant ions were selected for MS/MS. Data were searched using a local copy of Mascot searching against a Swissprot Mouse database (forward and reverse appended with common contaminants and BRCA1-C61G). The peptide tolerance was set to 10 ppm and the fragment ion tolerance was set to 0.02 Da. A maximum number of two missed cleavages by trypsin were allowed and carbamidomethylated cysteine and oxidised methionine were set as fixed and variable modifications, respectively.

### Western blotting
For a full list of antibodies, see Supplementary Table 1. Samples were run on SDS–PAGE protein gels and transferred to an Immobilon-P PVDF membrane. Following the transfer, membranes were blocked in 5% Marvel milk in PBS containing 0.1% Tween (PBStw), or in 5% BSA with PBStw, for 1 h before incubation with primary antibody at 4 °C for

16 h. Blots were washed in PBStw and then transferred into secondary horseradish peroxidase (HRP)-conjugated antibodies in 5% Marvel milk for 1 h. Blots were washed in PBStw before probing with 1:1 EZ-ECL mix (Biological Industries). Blots were exposed to X-ray film (WolfLabs) and developed using the Xograph Compact X4 developer. Densitometry calculations were performed where appropriate relative to loading controls using ImageJ[88].

### Immunofluorescence staining

Cells were plated at a density of $5 \times 10^4$ cells mL$^{-1}$ in 24-well plates on circular glass coverslips (13-mm diameter). Cells were then treated as described. For foci analysis cells were also subject to EdU-staining, these were incubated with EdU at a final concentration of 10 μM for 10 min before fixing and staining was carried out as detailed in the Click-iT EdU Imaging Kits (Life Technologies). Cells were pre-extracted by incubation with ice-cold 0.5% Triton X-100 in PBS on ice for 5 min before fixation with 4% PFA. Once fixed the cells were permeabilised for a further 30 min using 0.5 % Triton X-100 in PBS before incubation with a blocking solution (10% FCS in PBS for 30 min). Cells were then incubated with primary antibodies in 10% FCS in PBS at 4 °C overnight. The following day, cells were washed 3x with PBST before incubation with Fluorescent secondary antibody (1:2,000) for 2 h. Cells were then washed three times in PBST and the DNA was stained using Hoechst at a 1:20,000 concentration for 5 min. Excess Hoechst was removed by washing with PBS and coverslips were mounted onto Snowcoat slides using Immunomount mounting medium. For a full list of antibodies, see Supplementary Table 1. Immunofluorescence staining was imaged using a Leica DM6000B microscope with an HBO lamp with a 100-W mercury short-arc UV bulb and four filter cubes, A4, L5, N3 and Y5, which produce excitations at wavelengths 360, 488, 555 and 647 nm, respectively.

### Proximity linked ligation assay

MEFs were seeded at $4 \times 10^4$ cells ml$^{-1}$ onto poly-l-lysine-coated coverslips and irradiated with 2 Gy, before recovery for 3 h. Cells were pre-extracted for 5 min on ice with pre-extraction buffer (20 mM NaCl, 3 mM MgCl$_2$, 300 mM sucrose, 10 mM PIPES, 0.5% Triton X-100) and fixed in 4 % PFA for 10 min before blocking in 5% BSA for 16 h. Blocking medium was removed, and cells were then incubated with the primary antibodies in 5% FCS in PBS for 1 h at room temperature. After incubation with primary antibodies, cells were washed 2 × 5 min in wash buffer A (Sigma) and subsequently incubated with the MINUS or PLUS PLA probes (Sigma Duolink PLA kit) for 1 h at 37 °C in a warm foil-covered box. Cells were then washed twice for 5 min with wash buffer A (Sigma Duolink PLA kit) and incubated with the Sigma Duolink ligation kit (1× ligation buffer, ligase enzyme) for 30 min at 37 °C. Cells were washed twice for 5 min with wash buffer A and incubated for 100 min at 37 °C with the Sigma Duolink amplification kit (1× amplification buffer, polymerase enzyme). Subsequently, the cells were washed for 10 min with wash buffer B (Sigma Duolink PLA kit) at room temperature, incubated 5 min with Hoechst and washed again with wash buffer B twice for 5 minutes. Finally, cells were washed for 1 min with 0.01% wash buffer B and coverslips were mounted onto Snowcoat slides using Immunomount mounting medium. PLA dots were counted using a Leica DM6000B microscope with a HBO lamp with a 100-W mercury short-arc UV bulb and four filter cubes, A4, L5, N3 and Y5, which produce excitations at wavelengths 360, 488, 555 and 647 nm, respectively.

### Colony survival assays

Cells were plated in 24-well plates at $4 \times 10^4$ cells per ml and treated according to the experiment performed. Cells were trypsinised and plated in 6-well or 24-well plates at limiting dilutions followed by incubation for 5 days at 37 °C at 5 % CO$_2$. Once colonies had grown they were stained with 1% methylene blue in 50% ethanol or 0.5% crystal violet in 50% methanol and counted. For a full list of DNA-damaging agents and inhibitors, see Supplementary Table 4. Each experiment was normalised to untreated controls.

### Generation of stable cell lines

Stable cell lines were generated from Flp-In U2OS cells that were co-transfected with human BRCA1 cDNA variant in the pcDNA5/FRT/TO vector, and with the Flp recombinase cDNA in the pOG44 vector. Control transfections were carried out without the pOG44 recombinase. Two days after transfection, cells were selected with 100 μg/ mL hygromycin, the culture medium was replaced every 2 to 3 days and cells were selected for approximately 2 weeks. After selection, cells were treated with 2 μg/mL doxycycline for 72 h to induce expression of exogenous Flag-eGFP-BRCA1.

### Plasmid and siRNA transfection

FuGENE 6 (Roche) was used as a reagent to transfect cells with DNA plasmids. The ratio used was 4:1 FuGENE (μIL:DNA (μg), following the manufacturer's guidelines. siRNA transfections were carried out using the transfection reagent Dharmafect1 (Dharmacon) following the manufacturer's instructions. For a full list of siRNA sequences see Supplementary Table 2.

### Retrovirus production and infection

HEK293T Platinum E cells were transfected with pMSCV-IRES-GFP containing different BARD1 variants or RPA fusions using FuGENE-6 following the manufacturer's instructions. Culture media was collected 60 h later and filtered through a 0.45 mm filter. MEFs were infected with retroviral-containing media supplemented with 4 μg/ mL polybrene (Sigma). The RPA-70 constructs were generated after previously described BRCA2 fusion approaches[72]. In NLS-Ex27-RPA-flag: BRCA2$_{Exon 27}$ corresponds to ALDFLSRLPLPPPVSPICTFV-SPAAQKAFQPPRSCG (human BRCA2 residues 3,270-3,305). In NLS-BRC4-RPA70-Flag BRC4 corresponds to EKIKEPTLLGFHTASGKKVK IAKESLDKVKNLFDE (human BRCA2 residues 1,514-1,548).

### Fibre labelling and spreading

Cells were seeded in 6 cm$^2$ plates and treated with thymidine analogues CldU and IdU. To monitor fork protection, cells were incubated at 37 °C with 25 μM CldU for 20 min, followed by incubation with 250 μM IdU for 20 min and then with 5 mM HU for 3 h. Some fork protection experiments were performed with a singular CldU pulse for 20 mins followed by treatment with 5 mM HU for 3 hours. To monitor replication fork restart, cells were incubated at 37 °C with CldU for 20 min and then with 1 mM HU for 1 h. The HU was subsequently washed out with PBS and cells were incubated for a further 40 min in media containing 250 μM IdU at 37 °C.

After incubation with thymidine analogues, cells were washed twice with ice-cold PBS for 5 min trypsinised, and resuspended to a final concentration of $50 \times 10^4$ cells/mL PBS. Fibres spreads were prepared using spreading buffer (200 mM Tris pH7.4, 50 mM EDTA, 0.5 % SDS) and fixed in Methanol: Acetic acid (3:1). Fixed DNA spreads were stored at 4 °C until staining.

For fibre staining, DNA spreads were denatured with 2.5 M HCl for 1 h 15 min followed by blocking with 1% BSA/0.1% Tween20 in PBS for 1 h. Thymidine analogues were stained using Rat αBrdU antibody (Abcam, 1:2,000) to detect CldU and Mouse αBrdU antibody (Becton Dickinson, 1:500) to detect IdU. Primary antibodies were fixed with 4% PFA for 10 min prior to the addition of secondary antibodies (α-Rat AlexaFluor 555 and α-Mouse AlexaFluor 488, 1:500 each) for 2 h. Slides mounted with immunomount were kept at −20 °C until microscope analysis.

### S1 nuclease-modified fibre assay

Cells were seeded in 6 cm$^2$ plates and treated with 25 μM CldU for 20 min and 250 μM IdU for 40 min as described for unmodified fibre

labelling above. Co-treatment with Polθ inhibitor ART558 was as indicated for some experiments. Cells were subsequently permeabilized with CSK100 buffer (100 mM NaCl, 10 mM MOPS, 3 mM MgCl₂, 300 mM sucrose, 0.5% triton X-100, pH 7.0) for 10 min either directly after pulse-labelling or after 16 to 24 h release into fresh growth media containing 200 ng/mL nocodazole for the detection of ssDNA gaps in G2/M phase of the cell cycle. Nuclei were treated with either 20 U/mL S1 endonuclease (Invitrogen 18001016) to induce DSBs at sites of DNA gaps or mock-treated (S1 buffer: 30 mM sodium acetate, 10 mM zinc acetate, 5% glycerol, 50 mM NaCl, pH 4.6) for 30 min at 37 °C. Nuclei were harvested by scraping and DNA fibres spreads were prepared, stained, and analysed as described for the unmodified fibre assay.

### Single-molecule analysis of resection tracks (SMART) assay
Cells were treated with Polθ siRNA, Rad52i, and/or Mre11i mirin as indicated in the presence of 20 μM BrdU for the last 48 h to label the whole genome with thymidine analogue. DNA was harvested and spread as described for the unmodified DNA fibre assay above. Native fibre spreads were stained with mouse anti-BrdU (1:500) for 1.5 h, fixed with 4% PFA, and incubated with anti-mouse AlexaFluor 488 for 1.5 h.

### Post-replication repair (PRR) assay
Cells were incubated at 37 °C with 25 μM IdU for 1 h. Subsequently, cells were washed twice with PBS and placed in fresh growth media containing 200 ng/mL nocodazole for 16–24 h. During the final 4 h of nocodazole incubation, 20 μM CldU was added to the growth media to be incorporated during PRR (gap filling). Treatments (times and concentrations) for each experiment are indicated in each figure. In the case of siRNA treatment, cells were transfected 72 h prior to harvesting. DNA fibre spreads were prepared and stained as described for unmodified fibre labelling above with the exception that secondary antibodies were interchanged to allow for easier analysis of green CldU repair dots on red IdU stretches (α-Rat AlexaFluor 488 and α-Mouse AlexaFluor 555).

### DNA fibre analysis
For the quantification of fork protection, the length of bi-labelled CldU and IdU tracks were measured using ImageJ[3], and IdU/CldU ratios of these arbitrary lengths were plotted to assess IdU label shortening. Shift to lower IdU/CldU ratios is indicative of loss of fork protection. Fork stalling was assessed by scoring CldU-only labelled structures as stalled forks. Stalling was quantified by scoring the percentage of CldU-only fibres of all red-labelled structures. To quantify PPR events, minimum 10 events per sample were scored. Only the IdU tracts with centred CldU were taken into account. The length of IdU tract with at least one CldU dot was measured using ImageJ and the pixel values were converted into μm. Lengths were further converted into kilobases (1 μm = 2.59 kilobases)[89] and PRR density was calculated by dividing the total number of CldU dots on an IdU tract by the length of that IdU tract in kilobases. To quantify ssDNA gap formation in nascent DNA, IdU/CldU ratios of arbitrary lengths of bi-labelled structures were used to assess S1-dependent shortening of the IdU track. Shift to lower IdU/CldU ratios between mock- and S1-treated samples is indicative of ssDNA gap formation in nascent DNA. Gross ssDNA was assessed by measuring the lengths of green labelled native DNA patches. Arbitrary lengths were converted into μm using the scale bars created by the microscope.

### Metaphase spreads
MEFs were treated with 5 mM HU for 3 h and then incubated with colcemid (0.01 μg/mL) for 16 h. Cells were trypsinised and centrifuged at 300 g for 5 min. The supernatant was discarded, and cells were resuspended in PBS and centrifugated again. Five millilitres of ice-cold 0.56% KCl solution was added, and cells were incubated at 37 °C for

15 min before centrifuging at 300 g for 5 min. The supernatant was discarded, and the cell pellet was broken before fixation in 5 mL of ice-cold methanol: glacial acetic acid (3:1). Excess of fixation agents were removed and 10 μL of the cell suspension was dropped onto an acetic-acid-humidified slide. Slides were allowed to dry for at least 24 h and then stained with Giemsa solution (Sigma) diluted 1:20 for 20 min. Slide mounting was performed with Eukitt (Sigma).

### CRISPR/Cas9 HR assay
Adapted from[90]. MEFs (2 × 10⁶/condition) were electroporated using the 100 μL Neon electroporation system (1350 V, 30 ms, 1 pulse) to introduce 10 μg of pX459 V2.0 containing Cas9 and a gRNA targeting Rosa26 locus, alongside 10 μg of pUC57 containing a Rosa26 HR template with a 4 bp edited sequence. Following electroporation, cells were plated into antibiotic free media and allowed to recover. Cells were harvested 72 h later, and genomic DNA isolated using a DNEasy Blood and Tissue kit, following manufacturer's instructions. PCR was performed using GoTaq Green 2x master mix (Promega), and results analysed using agarose gel electrophoresis. HR specific Band intensities were quantified using ImageJ, and normalised to a HR-independent PCR product at the Rosa26 locus.

Alternatively, PCR was performed using PfU DNA Polymerase (Promega), and products were purified using AMPure XP magnetic beads (Beckman Coulter) following manufacturer's instructions. PCR products were barcoded, pooled and sequenced using a LSK109 library preparation kit on a single R.9.4.1 MinION flowcell (Oxford Nanopore Technologies) which was run for 4 h. Raw FAST5 files were base called with Guppy 5 to produce raw FASTQ files. These files then underwent read correction using Canu 2.2[91] using the –correctReads parameter. Reads were aligned to the mm10 mouse reference genome using minimap2 (version 2.24[92]) using the parameters: *ax map-ont*. CRISPResso2 (https://crispresso.pinellolab.partners.org/submission)[93] was then run targeting the Rosa26 locus of the mm10 genome. Each read was assigned as an HR outcome if matching the template sequence, a TMEJ outcome if matching predicted TMEJ sequences, or an NHEJ outcome if containing non-TMEJ indels. TMEJ predictions were carried out using MEDJED (http://www.genesculpt.org/medjed/). For a full list of primers, template and gRNA sequences see Supplementary Table 3.

### Statistics and reproducibility
All statistical tests used are indicated in the figure legend. All experiments were repeated at least once, and the number of biological replicates is reported for each experiment. To aid readability, statistics have only been shown between pertinent groups in figures.

### Reporting summary
Further information on research design is available in the Nature Portfolio Reporting Summary linked to this article.

## Data availability
All data generated in the study are included in this published article, including supplementary figures, or are available from the authors upon request. Source data are provided with this paper, and are available at figshare [https://doi.org/10.6084/m9.figshare.24270799].

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

## Acknowledgements

ART558 was a kind gift of Graeme Smith Artios Pharma Ltd. Cambridge, U.K. SV40 1: pBSSVD2005 was a gift from David Ron (Addgene plasmid # 21826; http://n2t.net/addgene:21826; RRID:Addgene_21826). Anti murine BARD1 antibody was a gift from Richard Baer (Columbia). We thank Claudia Lukas (University of Copenhagen) for the RAD52 antibody. 53BP1 animals were a gift from Andre Nussenzweig, NIH (National Institutes of Health), Bethesda. G.R., J.B. and K.S. are funded by Cancer research, UK (C8820/A28283), M.E. and K.S. by Breast Cancer Now, UK (2015MayPR499), E.A. by the Midlands Integrative Biosciences Training Partnership, funded by the Biotechnology and Biological Sciences Research Council, UK. A.J.G. by the Wellcome Trust (206343/Z/17/Z), A.B. and C.W. by Cancer Research UK (C31641/A23923). We thank the microscopy and Imaging services (MISBU) at the University of Birmingham Technology Hub for their assistance and maintenance of equipment. We sincerely apologise to colleagues whose excellent and relevant work was not referenced due to space restrictions.

## Author contributions

K.S. (colony assays, immunofluorescence, RAD51 foci analysis, metaphase spreads, DNA fibre assays, PRR assay, manuscript assembly and data management), G.R. (Mass Spec preparations, immunofluorescence, HR measurements, construct design RAD52, hBRCA1, colony assays and western blots), E.A. (PRR assay, colony assays, immunofluorescence and western blots), A.L.P. (S1-modified and SMART assays), L.C. (murine BARD1 constructs), A.J.G. (mouse colony), M.E. (antibody development, MEF generation), A.B. & C.W. (sequencing and bioinformatic analysis), J.B. (Genotyping), J.R.M. (supervision, project direction and manuscript writing). All authors read and reviewed the manuscript.

## Competing interests

The authors declare no competing interests.
