## [Peer Review File · Nature Communications]

Mechanisms of Synthetic Lethality Between BRCA1/2 and 53BP1 Deficiencies and DNA Polymerase Theta TargetingREVIEWER COMMENTS

Reviewer #1 (Remarks to the Author):

In this work Starowicz and colleagues dissect the complexity of DNA double strand break (DSB) repair dysregulatory pathways in tumors with BRCA1 mutation. In particular, the authors beautifully carve out the hierarchy of functional interactions between homologous recombination (HR) components BRCA1, BARD1, BRCA2 and the – simplified speaking - competing pathways towards single-strand annealing (SSA) involving RAD52 and microhomology-mediated end joining (MMEJ) relying on polymerase theta (POL θ). To this end the authors engage all thinkable genetic (knockout, knockin, knockdown, point mutants, etc.), cell biological (e.g. immunofluorescence microscopy-based colocalization, proximity ligation, fiber spreading assay) and biochemical (immunoprecipitation and mass spectrometry) technologies as well as an elegant DSB repair assay based on CRISPR/Cas9 cleavage and nanopore sequencing. This work considers combinations of multifactorial, often subtle interactions by combined analysis of mono- and biallelic changes of one pathway and modulation of protein levels or activities (different inhibitor concentrations) affecting other pathways, thereby more closely mimicking the situation in tumors. The analyses of mutations or fused peptides disrupting specific interactions and biochemical activities as for AAE-BARD1 or RPA-BRC4 are truly enlightening. Altogether, the manuscript sheds light on the complexity of the interplay between the targets of critical inhibitors of utmost interest for new developments of synthetic lethal strategies.

For further improvement of this work, I have three major comments and several minor ones.

Major comments:

- 1.) This manuscript is extremely dense and should be complemented by a graphical model summing up the multiple upstream and downstream roles of all the different players investigated and the impact of inhibitory drugs thereupon.
- 2.) In Figure 3e the authors make the interesting observation that only low RAD52 inhibitor concentrations rescue survival after Pol θ knockdown. However, the authors do not discuss this remarkable concentration dependency.
- 3.) The authors chose to engage transformed murine embryonic fibroblasts (MEFs) with the Brca1C61G mutation as a starting point for further modifications of the genetic background, and in this way aim at elucidating regulatory networks of relevance for treatment strategies of cancer patients. The manuscript categorizes the Brca1C61G mutation as hypomorphic, yet in recent efforts to classify human BRCA1 mutations by functional assays, BRCA1 p.Cys61Gly was found to be deleterious and classified as pathogenic (see e.g. Findlay et al. 2018 Nature). The manuscript text should contain such critical information on the human phenotype. Given some limitations of mouse models for human cancer with DSB repair dysfunction (see e.g. Banuelos et al. 2008 DNA Repair; Majhi et al. 2021 Oncogene), it will increase the impact of their findings substantially if the authors recapitulate key results on the BRCA1/53BP1-RAD52-POL θ hierarchy in human cells (so far only human cell data on BARD1 in Supplemental Fig. 5).

Minor comments:

- 1.) The statement on the ‘HR deficit of 24%’ in Fig. 1d is not based on mean/median values from replicates including statistics, why either quantitative PCR data should be generated as in Supplemental Figure 1j or the statement made more carefully.
- 2.) It is not clear to the reader, why it says that in ‘Brca1C61G/C61G 53bp1-/- cells we found that RNF168 depletion had a limited impact on BARD1 foci but substantially reduced numbers of RAD51’. Both foci changes are highly statistically significant in Figure 1f and Supplemental Figure 2b. When calculating relative changes in percentages the reduction might be similar. Please modify/delete the statement accordingly.
- 3.) Please correct the statement ‘Pol θ siRNA treatment had negligible impact on RNF168 foci formed at γ H2AX-decorated sites (Fig. 3a & b)’ to ‘Pol θ siRNA treatment had negligible impact on RNF168 and γ H2AX foci (Fig. 3a & b)’ as colocalization is not shown.

4.) Rad52 foci detection seemingly relied on detection of exogenous flag-tagged Rad52, which should be mentioned in the main text and described in the Methods section.

5.) Rad51 foci data reveal that the Rad51 foci accumulation kinetics change upon interference such as by Pol θ knockdown (Figure 2d), so that the differences in the kinetics or both time points should be discussed rather than one single time point. This also applies to Supplemental Figure 4g. In the legend to Figures 2c, 3f, g, it is not mentioned whether cells were treated and if so which time point post-treatment was chosen.

6.) Please apply the nomenclature for human versus murine genes and proteins, e.g. BRCA1 and BRCA1 versus Brca1 and Brca1, respectively.

7.) Schematically draw the fiber assay protocol above each panel showing fiber data. In the methods section three protocols named 'CldU fibre length', 'stability of nascent DNA' and 'restart' are described. The legend and the scheme above the panel should correspond to these descriptions in each case. From the description in the Methods it is not clear whether in Supplemental Figure 4d y-axis labeling is twisted (IdU/CldU ratio versus CldU/IdU ratio). The method for counting stalled forks in Supplemental Figure 4e and f is not clear, please specify in the methods section, legend and the scheme above the panels (e.g. asymmetry, ratio).

8.) Figure presentations:
Please provide the full genotype of the cells in each figure below each panel for the reader to not get lost, e.g. always indicate 53bp1 and Brca1 status even if the same for the whole panel. Equally, indicate the type of treatment in each figure panel (e.g. IR in Figure 1b. Given that the time point of foci detection matters (see e.g. Figure 2d), always indicate the time point of analysis post-treatment or post-start of treatment in the legend.
Representative microscopic images should be provided for each new protein analyzed such as for Rad51 in Figure 1b or Rnf168 in Figure 3b.
Show bee swarm graphs, whenever showing foci data, i.e. do not switch between columns and bee swarms such as in Figure 2c and d.
Switch the panels in Figure 1e and f as well as Supplemental Figure 1e and f according to the text flow.
In Figure 4 provide schemes indicating the positioning of mutations and peptides relative to the full-length BARD1 and BRCA2 proteins with functional domains.

9.) Statistics:
In the Materials and Methods section it says that statistics have only been shown between 'pertinent' groups in figures. Yet, for reasons of full transparency all statistically significant differences should be disclosed directly in the figure panel or at least in supplemental tables.
SEM may not be calculated from $n < 3$, why SD must be shown in all these cases.
Supplemental Figure 4b does not reveal any statistically significant differences, not even for the +ART558 positive control. Either new experimental data with more replicates should be provided or the corresponding figure panel and connected statement deleted. The same applies to Supplemental Figure 6c.

Reviewer #2 (Remarks to the Author):

Starowicz, Ronson et al studied the underlying cause of the synthetic-lethality relationship between BRCA1/2 and 53BP1 deficiencies and DNA polymerase Theta loss. The synthetically lethal relationship in cancers is currently an important avenue of therapeutics development. Therefore, the study may potentially have a significant impact. The main novel observation of this study is that the sensitivity of BRCA- and 53BP1-deficient cells to Pol Theta repression depends on RAD52. This result is somewhat unexpected as Pol θ and RAD52 are thought to belong to different DSB repair pathways, and therefore their inhibition may be expected to suppress the viability of BRCA-deficient cells in a synergistic manner. Indeed, contrary to the current report a previously published paper by Feng et al showed that RAD52 deletion contributes to lethality in POLQ-deficient cells. One possible explanation for this discrepancy is that the mechanisms of DSB repair are not the same in human and in murine cells that Starowicz, Ronson et al used in the current study. And the authors admit that their observations may

not be universal, which unfortunately significantly reduces the value of this study and its potential therapeutic implications. Thus, additional experiments in different cell lines, preferably human, are required.

More importantly, the authors conclusions leave some room for doubts. The key authors' conclusions are based on experiments with RAD52 inhibitors, particularly 6-OH-Dopamine (6-OHD). For instance, in the presence of 6-OHD Fig 3i shows an increase of cell viability; Fig. 3j shows a decrease in chromosomal aberrations; figure 4l shows an increase in survival of cells depleted for Pol Theta and BRCA2. It is also used in Supplementary Fig. 4 to demonstrate the effect RAD52 inhibition on replication fork progression and on survival of cells depleted for BRCA1 and treated with Pol Theta inhibitors.

There are several problems with this extensive usage of a small molecule inhibitor. First, small molecule inhibitors are known to have off-target effects; and 6-OHD was detected as a hit in numerous screening campaigns. Second, 6-OHD shows a very unusual behavior in the current study: its effects are increasing with the decrease in concentration. The maximal effect of 6-OHD was observed at a very low concentration: 0.15 μ M. In the original publication, when 6-OHD was introduced as a RAD52 inhibitor, it showed activity against RAD52 in cells at concentrations from 5 to 20 μ M, up to two orders of magnitude higher than in the current paper. It is possible that murine cells are more responsive to 6-OHD than human cells, in which it was initially characterized. However, Fig. 1h shows that 6-OHD decreases the survival of murine BRCA1 CG/CG cells defective in 53BP1 at 4-5 μ M, similar to human cells. Then, it is unclear what 6-OHD actually targets in these experiments, RAD52 or some unknown protein(s). Third, in Supplementary Fig. 4f, 6-OHD actually increases the fraction of stalled forks. This result that contradicts the authors' theory, is presented without comments. Similarly, there were no comments on result in Fig. 3i, where 6-OHD at concentration 1.25 μ M showed some decrease in cell survival. The bottom line: all the results with 6-OHD mentioned here should be reproduced using alternative approaches, e.g., siRAD52.

The authors do not discuss the underlying mechanism of RAD52-dependent sensitization of BRCA- and 53BP1 defective cells in which Pol Theta was suppressed. This a very counterintuitive result. How could inhibiting of RAD52 that is responsible for the only remaining DSB repair pathway in these cells, at least at the current state of knowledge, increase cell survival? In summary, a major revision is required before the paper can be considered for publication.

Specific comments:

1. Page 2. "However, while proteins involved in HR directly suppress TMEJ, it is not clear if the reverse is true."

What is the meaning of this sentence: Does it mean that in the reverse TMEJ suppress HR proteins or that TMEJ proteins suppress HR? In addition, several lines above the authors cited refs 2, 3, 6 that show that Pol θ suppresses HR. Clarification is needed.

2. In Fig. 1e, siRNF168 reduced survival of both CG/+ and CG/CG cells in the presence of 53bp $^{-/-}$. No explanation is provided for the sensitivity of the BRCA heterozygote, which is used in several instances as a control for CG/CG cells.

3. Reference 22 is cited for RNF168 promoting recruitment of BRCA1-BARD1, however, their own data do not show the effect of RNF168 depletion on BARD1 foci. What is the cause of the discrepancy?

4. Fig. 3j lacs 6-OHD alone.

5. Sup 4g. Control without siPol θ but with siRAD52 is missing.

6. While BRC4 may promote loading of RAD51 on RPA-covered ssDNA, no evidence was presented that this activity is conserved in the RPA-BRC4 construct.

Reviewer #3 (Remarks to the Author):

To the authors:

Recent studies have demonstrated that double knockout of BRCA1 and 53BP1 results in PARPi increased DSB end resection, at least some other features of HR repair, and PARP1i resistance.

However, these double k/o cells apparently retain their sensitivity to the POL θ inhibitor, ART558. In the current study, Starowicz et al examine the possible mechanism of synthetic lethality of Brca1(C61G/C61G), 53BP1(-/-) cells (here referred to as dko cells) with POL θ inhibitor.

Previous studies have shown that DKO cells have increased DSB end resection, accounting for increased RAD51 loading and improved HR repair. Knockout or inhibition of POL θ results in a further increase in DSB end resection and pRPA loading on the generated ssDNA. This is one known mechanism of SL, similar to the SL observed by a double knockout of a SHLD gene and POL θ .

Major Concerns:

1. The current study proposes an additional, and perhaps alternative, mechanism of Synthetic Lethality- namely, the increased in chromatin recruitment of RAD52. Overall, this concept is new, but some experimental evidence is lacking. Are these two mechanisms contradictory or additive? Or, are the mechanisms in fact part of the same process? Indeed, in their last paragraph, the authors point out that increased DSB end resection may account for the increased RAD52 recruitment. But they do not perform experiment so definitively prove this.

2. The paper does not identify any new protein players in this proposed mechanism of SL, and the story therefore seems somewhat weak for a Nature journal.

3. Figure 1 confirms that the dko cells have some elevation of HR repair. How does this increase compare to the HR in cells from wildtype murine cells from the same colony?

4. Figure 2 shows that Brca1 loss and 53BP1 loss results in elevated POL θ protein expression. Is this elevation due to the increase in resection at DSBs?

5. Figure 3 shows that the treatment of 53BP1(-/-) cells with POL θ inhibitor results in increased recruitment of RAD52 to foci and ultimately to cell death. The increase may be stronger in the dko cells. Is the increase in cell death result from a process of apoptosis (caspase cleavage). And does RAD52 knockdown or inhibition reduce the apoptosis? Does reduction in DSB end resection also reduce cell death? Are the two processes additive or synergistic?

6. How do you propose that RAD52 accumulation is toxic to the POL θ inhibitor treated dko cells?

Other Points:

1. Page 2. The authors claim that there is no evidence that proteins involved in TMEJ can suppress HR. This is not correct. For instance, POL θ itself has anti-recombinase activity and can remove RAD51 from sites of HR repair.

2. The paper does have some interesting translational implications. For instance, low levels or declining levels of RAD52 may be a predictive biomarker of developing POL θ inhibitor resistance, especially in cells with an underlying dko of Brca1 and 53BP1. Also, in some settings, the combination of a POL θ inhibitor and a RAD52 inhibitor would be expected to be antagonistic.

3. Do other processes of DSB increased resection, such as RIF1 or Shieldin complex depletion, also work with POL θ inhibition to further increase end resection and RAD52 loading? Does RAD52 knockdown also reduce the SL in this setting?

Response to reviewers' comments.

Many thanks to the reviewers for their insights and effort in reviewing our work. We have responded with point-by-point answers below.

Reviewer #1 (Remarks to the Author):

In this work Starowicz and colleagues dissect the complexity of DNA double strand break (DSB) repair dysregulatory pathways in tumors with BRCA1 mutation. In particular, the authors beautifully carve out the hierarchy of functional interactions between homologous recombination (HR) components BRCA1, BARD1, BRCA2 and the – simplified speaking - competing pathways towards single-strand annealing (SSA) involving RAD52 and microhomology-mediated end joining (MMEJ) relying on polymerase theta (POL θ). To this end the authors engage all thinkable genetic (knockout, knockin, knockdown, point mutants, etc.), cell biological (e.g. immunofluorescence microscopy-based colocalization, proximity ligation, fiber spreading assay) and biochemical (immunoprecipitation and mass spectrometry) technologies as well as an elegant DSB repair assay based on CRISPR/Cas9 cleavage and nanopore sequencing. This work considers combinations of multifactorial, often subtle interactions by combined analysis of mono- and biallelic changes of one pathway and modulation of protein levels or activities (different inhibitor concentrations) affecting other pathways, thereby more closely mimicking the situation in tumors. The analyses of mutations or fused peptides disrupting specific interactions and biochemical activities as for AAE-BARD1 or RPA-BRC4 are truly enlightening. Altogether, the manuscript sheds light on the complexity of the interplay between the targets of critical inhibitors of utmost interest for new developments of synthetic lethal strategies.

For further improvement of this work, I have three major comments and several minor ones.

Major comments:

1.) This manuscript is extremely dense and should be complemented by a graphical model summing up the multiple upstream and downstream roles of all the different players investigated and the impact of inhibitory drugs thereupon.

A graphical model is shown (Fig 7).

2.) In Figure 3e the authors make the interesting observation that only low RAD52 inhibitor concentrations rescue survival after Pol θ knockdown. However, the authors do not discuss this remarkable concentration dependency.

Now discussed, and recapitulated by titrations of other components of the pathway.

3.) The authors chose to engage transformed murine embryonic fibroblasts (MEFs) with the Brca1C61G mutation as a starting point for further modifications of the genetic background, and in this way aim at elucidating regulatory networks of relevance for treatment strategies of cancer patients. The manuscript categorises the Brca1C61G mutation as hypomorphic, yet in recent efforts to classify human BRCA1 mutations by functional assays, BRCA1 p.Cys61Gly was found to be deleterious and classified as pathogenic (see e.g. Findlay et al. 2018 Nature). The manuscript text should contain such critical information on the human phenotype.

The nature of the C61G mutation is described in more detail as requested.

Given some limitations of mouse models for human cancer with DSB repair dysfunction (see e.g. Banuelos et al. 2008 DNA Repair; Majhi et al. 2021 Oncogene), it will increase the impact of their findings substantially if the authors recapitulate key results on the BRCA1/53BP1-RAD52-POL θ hierarchy in human cells (so far only human cell data on BARD1 in Supplemental Fig. 5).

Several of the key observations have been tested, and found to be recapitulated in human breast epithelial cell line CAL51.

Minor comments:

1.) The statement on the 'HR deficit of 24%' in Fig. 1d is not based on mean/median values from replicates including statistics, why either quantitative PCR data should be generated as in Supplemental Figure 1j or the statement made more carefully.

Agreed, the description was misleading. The quantification is now described more carefully.

2.) It is not clear to the reader, why it says that in 'Brca1C61G/C61G 53bp1^{-/-} cells we found that RNF168 depletion had a limited impact on BARD1 foci but substantially reduced numbers of RAD51'. Both foci changes are highly statistically significant in Figure 1f and Supplemental Figure 2b. When calculating relative changes in percentages the reduction might be similar. Please modify/delete the statement accordingly.

Agreed, the statement is now changed accordingly.

3.) Please correct the statement 'Pol θ siRNA treatment had negligible impact on RNF168 foci formed at γ H2AX-decorated sites (Fig. 3a & b)' to 'Pol θ siRNA treatment had negligible impact on RNF168 and γ H2AX foci (Fig. 3a & b)' as colocalisation is not shown.

Agreed, the statement is now changed accordingly.

4.) Rad52 foci detection seemingly relied on detection of exogenous flag-tagged Rad52, which should be mentioned in the main text and described in the Methods section.

Yes, now mentioned in both.

5.) Rad51 foci data reveal that the Rad51 foci accumulation kinetics change upon interference such as by Pol θ knockdown (Figure 2d), so that the differences in the kinetics or both time points should be discussed rather than one single time point. This also applies to Supplemental Figure 4g. In the legend to Figures 2c, 3f, g, it is not mentioned whether cells were treated and if so which time point post-treatment was chosen.

Agreed, it wasn't clear. The RAD51 foci kinetics experiment has been removed for space reasons.

6.) Please apply the nomenclature for human versus murine genes and proteins, e.g. BRCA1 and BRCA1 versus Brca1 and Brca1, respectively.

We have followed standard nomenclature guidelines (sources listed below). We agree with the reviewer that it makes the manuscript harder to follow and would happily use the reviewers' suggestion should the Nature Comms editor agree that an exception is reasonable in this case.

We followed guidelines from the Oxford University Press: "Mouse: same as the gene symbol, but not italicised and all upper case

https://academic.oup.com/molehr/pages/Gene_And_Protein_Nomenclature and the Jackson Laboratory: https://www.informatics.jax.org/mgihome/nomen/short_gene.shtml "**Protein**

designations: Protein designations follow the same rules as gene symbols, with the following two distinctions: "Protein symbols use all uppercase letters, Protein symbols are not italicised". HGNC recommends italics for symbols denoting genes, mRNAs, and alleles. Elsevier and JCI Style says: Names.aspx *Mice and rats:* Gene symbols are italicised, with only the first letter in upper-case (e.g., *Gfap*). Protein symbols are not italicised, and all letters are in upper-case (e.g., GFAP).

We note that recent Nature Comms papers follow these rules (see

<https://www.nature.com/articles/s41467-023-37617-3>, <https://www.nature.com/articles/s41467-023-35938-x> <https://www.nature.com/articles/s41586-023-05940-w> although there are exceptions: <https://www.nature.com/articles/s41598-023-31484-0>)

7.) Schematically draw the fiber assay protocol above each panel showing fiber data. In the methods section three protocols named 'CldU fibre length', 'stability of nascent DNA' and 'restart' are described. The legend and the scheme above the panel should correspond to these descriptions in each case. From the description in the Methods it is not clear whether in Supplemental Figure 4d y-axis labeling is twisted (IdU/CldU ratio versus CldU/IdU ratio). The method for counting stalled forks in Supplemental Figure 4e and f is not clear, please specify in the methods section, legend and the scheme above the panels (e.g. asymmetry, ratio).

Many thanks – schematics are now shown for each relevant experiment which improves the understanding.

8.) Figure presentations:

Please provide the full genotype of the cells in each figure below each panel for the reader to not get lost, e.g. always indicate 53bp1 and Brca1 status even if the same for the whole panel. Equally, indicate the type of treatment in each figure panel (e.g. IR in Figure 1b. Given that the time point of foci detection matters (see e.g. Figure 2d), always indicate the time point of analysis post-treatment or post-start of treatment in the legend.

Genotypes are now included as requested. All fixation time points are now given in legends.

Representative microscopic images should be provided for each new protein analysed such as for Rad51 in Figure 1b or Rnf168 in Figure 3b.

Show bee swarm graphs, whenever showing foci data, i.e. do not switch between columns and bee swarms such as in Figure 2c and d.

Switch the panels in Figure 1e and f as well as Supplemental Figure 1e and f according to the text flow.

In Figure 4 provide schemes indicating the positioning of mutations and peptides relative to the full-length BARD1 and BRCA2 proteins with functional domains.

All changes included, representative images are now in Supplemental Information, bee swarms given and position of mutations illustrated.

9.) Statistics:

In the Materials and Methods section it says that statistics have only been shown between 'pertinent' groups in figures. Yet, for reasons of full transparency all statistically significant differences should be disclosed directly in the figure panel or at least in supplemental tables. SEM may not be calculated from $n < 3$, why SD must be shown in all these cases.

Supplemental Figure 4b does not reveal any statistically significant differences, not even for the +ART558 positive control. Either new experimental data with more replicates should be provided or the corresponding figure panel and connected statement deleted. The same applies to Supplemental Figure 6c.

We have increased the statistical reporting within the figures and also included statistical tables with the source data files— we hope this strikes the balance between transparency and readability.

We thank the reviewer for the note on 4b/Sup 6c. Our revisiting of this data led to an important revelation as it showed these were not significant and that the RAD52 suppression (or RPA depletion of MRE11 inhibition) did not reverse the impact of the Pol-theta inhibitor (in contrast to Pol-theta depletion). Our follow-up experiments excitingly now reveal the different basis of sensitivity between *Brca1*^{C61G/C61G} *53bp1*^{-/-} cells and *53bp1*^{-/-} cells to the inhibitor.

Reviewer #2 (Remarks to the Author):

Reviewer #2 (Remarks to the Author):

Starowicz, Ronson et al studied the underlying cause of the synthetic-lethality relationship between BRCA1/2 and 53BP1 deficiencies and DNA polymerase Theta loss. The synthetically lethal relationship in cancers is currently an important avenue of therapeutics development. Therefore, the study may potentially have a significant impact.

1. The main novel observation of this study is that the sensitivity of BRCA- and 53BP1-deficient cells to Pol Theta repression depends on RAD52. This result is somewhat unexpected as Polθ and RAD52 are thought to belong to different DSB repair pathways, and therefore their inhibition may be expected to suppress the viability of BRCA-deficient cells in a synergistic manner. Indeed, contrary to the current report a previously published paper by Feng et al showed that RAD52 deletion contributes to lethality in POLQ-deficient cells. One possible explanation for this discrepancy is that the mechanisms of DSB repair are not the same in human and in murine cells that Starowicz, Ronson et al used in the current study. And the authors admit that their observations may not be universal, which unfortunately significantly reduces the value of this study and its potential therapeutic implications. Thus, additional experiments in different cell lines, preferably human, are required.

We apologise for the confusion we have created and have explained the concentration effect much more in the discussion. Indeed we find a high concentration of RAD52 inhibitor or siRNA is not compatible with Pol-Theta depletion (the most similar condition full knockout of each). Our findings are that low-level suppression of RAD52 (or RPA or MRE11) suppresses the sensitivity of BRCA1 and 53BP1 deficient cells to Pol-Theta depletion, without exposing the cells to the vulnerabilities of losing these components.

The Feng *et al* work mentioned also used murine cells (*Polq* knockout MEFs). As suggested, we tested human (breast epithelial) cells and found a recapitulation of our findings in this setting also.

2. More importantly, the authors conclusions leave some room for doubts. The key authors' conclusions are based on experiments with RAD52 inhibitors, particularly 6-OH-Dopamine (6-OHD). For instance, in the presence of 6-OHD Fig 3i shows an increase of cell viability; Fig. 3j shows a decrease in chromosomal aberrations; figure 4l shows an increase in survival of cells depleted for Pol Theta and BRCA2. It is also used in Supplementary Fig. 4 to demonstrate the effect RAD52 inhibition on replication fork progression and on survival of cells depleted for BRCA1 and treated with Pol Theta inhibitors.

There are several problems with this extensive usage of a small molecule inhibitor. First, small molecule inhibitors are known to have off-target effects; and 6-OHD was detected as a hit in numerous screening campaigns. Second, 6-OHD shows a very unusual behavior in the current study: its effects are increasing with the decrease in concentration. The maximal effect of 6-OHD was observed at a very low concentration: 0.15 uM. In the original publication, when 6-OHD was introduced as a RAD52 inhibitor, it showed activity against RAD52 in cells at concentrations from 5 to 20 uM, up to two orders of magnitude higher than in the current paper. It is possible that murine cells are more responsive to 6-OHD than human cells, in which it was initially characterised. However, Fig. 1h shows that 6-OHD decreases the survival of murine BRCA1 CG/CG cells defective in 53 BP1 at 4-5 uM, similar to human cells. Then, it is unclear what 6-OHD actually targets in these experiments, RAD52 or some unknown protein(s).

Similarly, there were no comments on result in Fig. 3i, where 6-OHD at concentration 1.25 uM showed some decrease in cell survival. The bottom line: all the results with 6-OHD mentioned here should be reproduced using alternative approaches, e.g., siRAD52.

We have taken several approaches to address these concerns:

- As suggested, we tested siRNA to RAD52 in human and mouse cell assays – we find these approaches recapitulate the findings using 6-OHD.
- We examined the impact of a different RAD52 inhibitor (D-103) – which also improves cell viability.
- We tested titrations of siRNA to RPA (required for RAD52 recruitment and binding to ssDNA) and recapitulated the finding.
- We tested a mutant of RAD52 unable to interact with RPA, finding that unlike WT-RAD52 its expression does not suppress cell survival in the presence of Pol-Theta siRNA.
- We find a 'bell-curve' relationship between RAD52 suppression and cell survival in Pol-theta siRNA-treated *Brca1*^{C61G/C61G} *53bp1*^{-/-} cells. The full titration data is shown below for the reviewer.

Brca1^{C61G/C61G} 53bp1^{-/-} cells treated with Non-targeting control siRNA (NTC) or Pol-theta siRNA “Q” and decreasing amounts of 6-OHD (μM)

Third, in Supplementary Fig. 4f, 6-OHD actually increases the fraction of stalled forks. This result that contradicts the authors’ theory, is presented without comments.

This data is now presented with comments, alongside several other cellular defects that are worsened by the addition of low-concentration 6-OHD and that do not correlate with cellular sensitivities. Importantly, our discovery of the robust relationship between Pol-theta and RAD52 in cell viability has allowed us to identify a cellular defect that closely correlates with that relationship – and thus is likely to identify the process where Pol-theta is most relevant to cell survival.

3. The authors do not discuss the underlying mechanism of RAD52-dependent sensitization of BRCA- and 53BP1 defective cells in which Pol Theta was suppressed. This is a very counterintuitive result. How could inhibiting of RAD52 that is responsible for the only remaining DSB repair pathway in these cells, at least at the current state of knowledge, increase cell survival? In summary, a major revision is required before the paper can be considered for publication.

We agree our findings are counter-intuitive and we greatly appreciate this criticism. We have performed many experiments to tackle this point and as a result, the revised manuscript is the major revision requested. After exploring many known aspects of Pol-theta and finding they do not correlate with the Pol-theta:RAD52 relationship in cell viability we have now identified a function of Pol-theta that was not previously known. The data indicate that Polθ suppresses RAD52-mediated inhibition of G2/M gap-filling in BRCA1- and 53BP1-deficient cells and that this closely correlates with the features of the Pol-theta:RAD52 relationship in cell viability.

Specific comments:

1. Page 2. “However, while proteins involved in HR directly suppress TMEJ, it is not clear if the reverse is true.”

What is the meaning of this sentence: Does it mean that in the reverse TMEJ suppress HR proteins or that TMEJ proteins suppress HR? In addition, several lines above the authors cited refs 2, 3, 6 that show that Polθ suppresses HR. Clarification is needed.

We apologise for the poor writing, now clarified.

2. In Fig. 1e, siRNF168 reduced survival of both CG/+ and CG/CG cells in the presence of 53bp-/. No explanation is provided for the sensitivity of the BRCA heterozygote, which is used in several instances as a control for CG/CG cells.

We have now explained the observation further in the introduction to the relevant section. Our findings recapitulate the 2019 report from the Nussenzweig lab showing the relationship between BRCA1 insufficiency and RNF168 dependence (DOI: 10.1016/j.molcel.2018.12.010)

3. Reference 22 is cited for RNF168 promoting recruitment of BRCA1-BARD1, however, their own data do not show the effect of RNF168 depletion on BARD1 foci. What is the cause of the discrepancy?

A statistically significant (albeit small) difference is seen. Now rewritten so to not be misleading.

4. Fig. 3j lacs 6-OHD alone.

Data now shown.

5. Sup 4g. Control without siPolθ but with siRAD52 is missing.

Data now shown.

6. While BRC4 may promote loading of RAD51 on RPA-covered ssDNA, no evidence was presented that this activity is conserved in the RPA-BRC4 construct.

In the revision we now demonstrate that the RPA-BRC4 construct can induce RAD51 in cells depleted for BRCA1, indicating activity.

Reviewer #3 (Remarks to the Author):

Recent studies have demonstrated that double knockout of BRCA1 and 53BP1 results in PARPi increased DSB end resection, at least some other features of HR repair, and PARP1i resistance. However, these double k/o cells apparently retain their sensitivity to the POLθ inhibitor, ART558. In the current study, Starowicz et al examine the possible mechanism of synthetic lethality of Brca1(C61G/C61G), 53BP1(-/-) cells (here referred to as dko cells) with POLθ inhibitor.

Previous studies have shown that DKO cells have increased DSB end resection, accounting for increased RAD51 loading and improved HR repair. Knockout or inhibition of POLθ results in a further increase in DSB end resection and pRPA loading on the generated ssDNA. This is one known mechanism of SL, similar to the SL observed by a double knockout of a SHLD gene and POLθ.

Major Concerns:

1. The current study proposes an additional, and perhaps alternative, mechanism of Synthetic Lethality- namely, the increased in chromatin recruitment of RAD52. Overall, this concept is new, but some experimental evidence is lacking. Are these two mechanisms contradictory or additive? Or, are the mechanisms in fact part of the same process? Indeed, in their last paragraph, the authors point out that increased DSB end resection may account for the increased RAD52 recruitment. But they do not perform experiment so definitively prove this.

We have performed several new experiments to answer these questions. In summary, we show *how* Pol-theta loss leads to increased resection –through RAD52 promotion of nuclease recruitment, and we show *where* – not at DSBs or nascent ssDNA gaps – but through loss of Pol-Thea mediated repression of RAD52 suppression of G2/M fill-in synthesis.

2. The paper does not identify any new protein players in this proposed mechanism of SL, and the story therefore seems somewhat weak for a Nature journal.

We appreciate the reviewer’s comment and have made considerable progress delving into the mechanistic aspects of our findings. In addition to the central role of RAD52 in mediating the synthetic-lethal relationship between BRCA1/2 loss, 53BP1 loss and Pol-Theta depletion of the first submission, we now additionally show for the first time:

- A new function for Pol-theta - in suppressing RAD52-mediated repression of DNA synthesis.
- That 53BP1 is required to promote late-phase fill-in DNA synthesis.
- That Pol-theta inhibition with ART558 has a differential impact on *Brca1^{C61G/C61G} 53bp1^{-/-}* cells and *53bp1^{-/-}* cells and what determines that difference.

These data present a new and unexpected role for Pol-theta and RAD52, making the revised story of the required novelty for a Nature journal.

3. Figure 1 confirms that the dko cells have some elevation of HR repair. How does this increase compare to the HR in cells from wildtype murine cells from the same colony?

It is increased, this data is now shown

4. Figure 2 shows that Brca1 loss and 53BP1 loss results in elevated POLθ protein expression. Is this elevation due to the increase in resection at DSBs?

We don’t know the regulatory mechanism of Pol-Theta expression and can't say if resection feeds back to the control of transcription or protein stability. It's an interesting and important question, but one we feel is outside the topic of this manuscript.

5. Figure 3 shows that the treatment of 53BP1(-/-) cells with POLθ inhibitor results in increased recruitment of RAD52 to foci and ultimately to cell death. The increase may be stronger in the dko cells. Is the increase in cell death result from a process of apoptosis (caspase cleavage). And does RAD52 knockdown or inhibition reduce the apoptosis? Does reduction in DSB end resection also reduce cell death? Are the two processes additive or synergistic?

We have followed up this query by addressing the potential link between RAD52 and resection, finding that RAD52 binding to RPA is needed for its toxicity, that RAD52 encourages MRE11:ssDNA interaction and that RAD52 inhibition suppresses extended ssDNA in Pol-theta depleted cells. Moreover, we have identified a new role for RAD52 finding it suppresses G2/M gap-filling DNA synthesis when Pol-Theta is depleted. We also confirm that inhibition of RAD52 correlates with improved gap-filling and suppression of DSBs (and suppression of chromosome breaks and micronuclei) and cell death. We haven't addressed how the cells die, and feel in light of the new findings to support how RAD52 functions, that the answer is tangential to the manuscript.

6. How do you propose that RAD52 accumulation is toxic to the POL θ inhibitor treated dko cells?

We propose, and have evidence to show, that RAD52 is toxic to Pol-theta-depleted double-mutant cells and to Pol-theta-inhibited 53BP1-deficient cells through its ability to suppress G2/M DNA-synthesis through the promotion of nuclease function. We don't discount (we have no evidence for or against) that a further function, such as inappropriate recombination, also contributes.

Remarkably we find that RAD52 suppression is not able to reverse the toxicity of the Pol-theta inhibitor in double-mutant cells unless Pol-theta is depleted, revealing a role in the double-mutant cells for the inhibited polymerase. We find resistance to the inhibitor can be restored by increased concentrations of RAD51-mediators – although we don't know yet whether these are effective because of their activity at fork junctions or at the G2/M fill-in.

Other Points:

1. Page 2. The authors claim that there is no evidence that proteins involved in TMEJ can suppress HR. This is not correct. For instance, POL θ itself has anti-recombinase activity and can remove RAD51 from sites of HR repair.

We did not mean to imply this and hope the rewrite is clearer.

2. The paper does have some interesting translational implications. For instance, low levels or declining levels of RAD52 may be a predictive biomarker of developing POL θ inhibitor resistance, especially in cells with an underlying dko of Brca1 and 53BP1. Also, in some settings, the combination of a POL θ inhibitor and a RAD52 inhibitor would be expected to be antagonistic.

Indeed. The new findings go further- indicating that inhibitor resistance mechanisms may differ between *Brca1*^{C61G/C61G} *53bp1*^{-/-} cells and *53bp1*^{-/-} cells.

3. Do other processes of DSB increased resection, such as RIF1 or Shieldin complex depletion, also work with POL θ inhibition to further increase end resection and RAD52 loading? Does RAD52 knockdown also reduce the SL in this setting?

Yes, the relationship holds in other settings of extended resection. We addressed this question by using USP48 knockdown, which increases resection in the presence of 53BP1-Shieldin. We find USP48-depleted cells are similarly sensitive to Pol-theta inhibitor, and the sensitivity is similarly reversed by co-treatment with RAD52 inhibitor.

Reviewers' comments:

Reviewer #1 (Remarks to the Author):

In the revised version of their manuscript the authors erased my concerns and satisfied my requests. But way beyond, I believe with the newly unraveled concentration dependency of action of RAD52 on POLtheta inhibition this work provides a novel and highly interesting mechanism that may play a significant role in tumor evolution and development of treatment resistance. This finding emphasizes the need for such types of non-digital studies better reflecting the complexity and dynamics of tumor biology.

Reviewer #2 (Remarks to the Author):

My previous major concern has not been adequately addressed in this revision. Specifically, I indicated that the authors used a neurotoxin 6-hydroxidopamine (6-OHD) at concentrations about 100-fold lower than that it is required for inhibition of the protein of interest, RAD52. In few experiments they also used RAD52 siRNA. But again, they used it in unconventionally low concentrations. Nonetheless, all their conclusions assume that RAD52 is inhibited under these conditions. Therefore, I believe that the observed effects are nonspecific, and the conclusions of this paper are grossly incorrect.

Also, RAD52 binding to ssDNA is RPA-independent, the references 44-46 were cited incorrectly.

Reviewer #3 (Remarks to the Author):

To the authors:

I do think that it is interesting that, in some genomic context, the knockdown or inhibition of RAD52 can result in cellular resistance to a POLQ inhibitor. But this major point of the paper is lost in the current version.

On a positive note, the authors did address my major criticism. They confirmed that 53BP1 knockout can increase DSB end resection, leading to increased RAD52 loading. And now they confirm that another knockout -namely the knockout of USP28, can also increase DSB end resection and increased RAD52 loading. That's interesting.

It is interesting that POLQ can suppress RAD52 activity. POLQ can suppress RAD52-mediated nuclease resection and can suppress DNA synthesis. And RAD52 does seem to suppress G2-M gap filling. But these are descriptive observations. The paper does not show how POLQ can suppress RAD52. And does not show how RAD52 can suppress G2-M gap filling. The molecular mechanisms for these suppressive activities is not shown.

Overall, I believe that are trying to do too many things in one paper in order to prove two distinct mechanisms. Accordingly, the major impact of the paper is completely lost, and there are now 86 references. The original version of this paper was dense. Now it is even more dense.

I have read the abstract of the paper many, many times, on separate occasions, and I cannot make any sense out of it. For instance, what does this sentence from the abstract really mean?
"Polθ suppresses RAD52-mediated inhibition of G2/M gap-filling in BRCA1-deficient and 53BP1-deficient cells and RPA:RAD52 encourages MRE11 nuclease accumulation, suppressing DNA synthesis"
This sentence is very difficult to interpret.

And at one point in the paper, the authors use five different inhibitors (Olaparib, mirin, ART558, 6-

OHD, and USP48 siRNA), to prove these two distinct mechanisms. Genetic arguments would be stronger than drug-related arguments.

In my opinion, the best way for the authors to prove their two distinct mechanisms is to write two separate papers – one describing mechanism 1, and one describing mechanism 2. Otherwise, a side-by-side comparison of these two mechanisms is not instructive, since the mechanisms are so different.

I wish I could be more positive.

REVIEWER COMMENTS

Reviewer #1 (Remarks to the Author):

Ad “concentrations about 100-fold lower that is required for inhibition of the protein of interest, Rad52”:

The authors deliberately titrated RAD52 inhibitor (6-OHD) down to 100-fold lower concentrations than previously used for inhibition of ssDNA binding *in vitro*, single-strand annealing (SSA) and survival in cells (Chandramouly et al 2015 Chem Biol). It was previously shown that absence of BRCA1 or BRCA2 reduces the IC50 value dramatically (Chandramouly et al 2015 Chem Biol), so that additional loss of 53bp1 (Fig. 1h) or on top inhibition of POLtheta (Fig. 3h) can explain the hypersensitivity seen here. Moreover, in previous work DNA replication (stress) was either not investigated (Chandramouly et al 2015 Chem Biol) or only at a single 6-OHD concentration (Malacaria et al 2019 Nat Commun). In our previous work (Volcic et al 2020 Nat Microbiol), we have demonstrated an epistatic role of RAD52 in HIV-1 Vpu-mediated regulation of homologous repair (homologous recombination and SSA). In this previous work we noticed that siRNA-mediated downregulation of RAD52 by as little as 30% can abolish any further effect by HIV 1 on homologous repair in human lymphoblasts suggesting that already small level changes significantly impact on some of RAD52’s biological functions. The same loss-of-effect was seen after pharmacological inhibition by 6-OHD (5uM). Back then we concluded that both manipulations might be detrimental to the formation of functional multimeric rings of RAD52 (Stasiak et al 2000 Curr Biol; Chandramouly et al 2015 Chem Biol). As discussed in the models by Chandramouly et al, it is conceivable that sub-stoichiometric concentrations of an inhibitor may affect certain functions of RAD52 relying on a specific oligomeric shape of RAD52 (heptameric, undecameric, super-structures, stacked) by a conformational change, while high concentrations may disrupt the multimeric structures or even affect activities of dimeric RAD52. Notably, oligomerization in the cytoplasm was reported to promote nuclear import due to the weak individual nuclear localization signals (reviewed in Toma et al 2019 Cancers). Further, RAD52 possesses two DNA-binding sites that can promote SSA or D-loop formation depending on the involvement of ssDNA only or ssDNA and double-stranded DNA (Kagawa et al 2008 JBC). Finally, RAD52 exerts a multitude of biochemical activities in RNA-templated repair, DNA repair and replication (reviewed in Jalan et al 2019 Cancers; Gottifredi and Wiesmüller 2020 Cancers). Therefore, we are far from understanding the complexity of RAD52 functions and their potentially synergistic regulation by multimerization and heterotypic protein and/or DNA interactions. Accordingly, different levels of functional RAD52 may very well promote different biochemical activities of RAD52 and/or biological outcomes.

Intriguingly, in two completely independent studies which are in the process of submission, we and collaborators similarly discovered that two other DNA repair and replication factors exert at first sight opposing activities at different cellular concentrations. The “non-digital” view presented here by Starowicz et al is therefore needed to truly understand the complex interplay of these key molecules rather than just making a snapshot such as after complete knockout. Stunningly, in our study, we successfully used exactly the same range of siRNA concentrations to gradually alter the level of the protein of interest as used by Starowicz et al. One possibility that I can see to further strengthen the manuscript could be to verify the gradual downregulation of RAD52 post-knockdown in titration analysis. As RAD52 antibodies are known to be notoriously difficult to be used in Western Blotting, the authors might try to engage qRT-PCR to this end.

Most importantly, RAD52 inhibitor and knockdown titration data match nicely in this manuscript by Starowicz et al, which makes it very unlikely that the same “nonspecific” effects occurred. In fact, you would expect more off-target effects by higher drug and siRNA concentrations than by lower ones.

Ad “In few experiments they also used siRNA”:

The authors used Rad52 inhibitor and siRNA directed against Rad52 side-by-side in many experiments, namely in Fig.1 h and j

Fig. 3e and h, 3f and i
Fig. 4k
Fig. 5b and c

Reviewer #4 (Remarks to the Author):

I have read the paper a few times now (as it is rather complicated!) and I am convinced with the way the authors deplete and inactivate RAD52.

The fact that both low concentrations and the siRNA and the catalytic inhibitor show exactly the same results in viability as well as different assays (RAD51 foci, breaks, gaps etc) used by the authors convinces me that the results are trustable and valid.

Reading through the manuscript and the answers to the reviewers I am aligning with their interpretations and the conclusions.

Reviewer #1 (Remarks to the Author):

Ad “concentrations about 100-fold lower that is required for inhibition of the protein of interest, Rad52”:

The authors deliberately titrated RAD52 inhibitor (6-OHD) down to 100-fold lower concentrations than previously used for inhibition of ssDNA binding in vitro, single-strand annealing (SSA) and survival in cells (Chandramouly et al 2015 Chem Biol). It was previously shown that absence of BRCA1 or BRCA2 reduces the IC50 value dramatically (Chandramouly et al 2015 Chem Biol), so that additional loss of 53bp1 (Fig. 1h) or on top inhibition of POLtheta (Fig. 3h) can explain the hypersensitivity seen here. Moreover, in previous work DNA replication (stress) was either not investigated (Chandramouly et al 2015 Chem Biol) or only at a single 6-OHD concentration (Malacaria et al 2019 Nat Commun).

In our previous work (Volcic et al 2020 Nat Microbiol), we have demonstrated an epistatic role of RAD52 in HIV-1 Vpu-mediated regulation of homologous repair (homologous recombination and SSA). In this previous work we noticed that siRNA-mediated downregulation of RAD52 by as little as 30% can abolish any further effect by HIV 1 on homologous repair in human lymphoblasts suggesting that already small level changes significantly impact on some of RAD52’s biological functions. The same loss-of-effect was seen after pharmacological inhibition by 6-OHD (5uM). Back then we concluded that both manipulations might be detrimental to the formation of functional multimeric rings of RAD52 (Stasiak et al 2000 Curr Biol; Chandramouly et al 2015 Chem Biol). As discussed in the models by Chandramouly et al, it is conceivable that sub-stoichiometric concentrations of an inhibitor may affect certain functions of RAD52 relying on a specific oligomeric shape of RAD52 (heptameric, undecameric, super-structures, stacked) by a conformational change, while high concentrations may disrupt the multimeric structures or even affect activities of dimeric RAD52. Notably, oligomerization in the cytoplasm was reported to promote nuclear import due to the weak individual nuclear localization signals (reviewed in Toma et al 2019 Cancers). Further, RAD52 possesses two DNA-binding sites that can promote SSA or D-loop formation depending on the involvement of ssDNA only or ssDNA and double-stranded DNA (Kagawa et al 2008 JBC). Finally, RAD52 exerts a multitude of biochemical activities in RNA-templated repair, DNA repair and replication (reviewed in Jalan et al 2019 Cancers; Gottifredi and Wiesmüller 2020 Cancers). Therefore, we are far from understanding the complexity of RAD52 functions and their potentially synergistic regulation by multimerization and heterotypic protein and/or DNA interactions. Accordingly, different levels of functional RAD52 may very well promote different biochemical activities of RAD52 and/or biological outcomes.

Intriguingly, in two completely independent studies which are in the process of submission, we and collaborators similarly discovered that two other DNA repair and replication factors exert at first sight opposing activities at different cellular concentrations. The “non-digital” view presented here by Starowicz et al is therefore needed to truly understand the complex interplay of these key molecules rather than just making a snapshot such as after complete knockout. Stunningly, in our study, we successfully used exactly the same range of siRNA concentrations to gradually alter the level of the protein of interest as used by Starowicz et al. One possibility that I can see to further strengthen the manuscript could be to verify the gradual downregulation of RAD52 post-knockdown in titration analysis. As RAD52 antibodies are known to be notoriously difficult to be used in Western Blotting, the authors might try to engage qRT-PCR to this end.

Most importantly, RAD52 inhibitor and knockdown titration data match nicely in this manuscript by Starowicz et al, which makes it very unlikely that the same “nonspecific” effects occurred. In fact,

you would expect more off-target effects by higher drug and siRNA concentrations than by lower ones.

Ad “In few experiments they also used siRNA”:

The authors used Rad52 inhibitor and siRNA directed against Rad52 side-by-side in many experiments, namely in

Fig.1 h and j

Fig. 3e and h, 3f and i

Fig. 4k

Fig. 5b and c

We sincerely thank the reviewer for a thorough analysis of both our paper and the RAD52 literature. In relation to showing the RAD52 impact (as the reviewer says westerns are tricky) we were nevertheless able to show partial knock-down at the protein level (Fig 3g).

Reviewer #3 (Remarks to the Author):

To the authors:

I do think that it is interesting that, in some genomic context, the knockdown or inhibition of RAD52 can result in cellular resistance to a POLQ inhibitor. But this major point of the paper is lost in the current version.

Many thanks for the comment – we have edited the abstract to make the point clearer.

On a positive note, the authors did address my major criticism. They confirmed that 53BP1 knockout can increase DSB end resection, leading to increased RAD52 loading. And now they confirm that another knockout -namely the knockout of USP28, can also increase DSB end resection and increased RAD52 loading. That’s interesting.

It is interesting that POLQ can suppress RAD52 activity. POLQ can suppress RAD52-mediated nuclease resection and can suppress DNA synthesis. And RAD52 does seem to suppress G2-M gap filling. But these are descriptive observations. The paper does not show how POLQ can suppress RAD52. And does not show how RAD52 can suppress G2-M gap filling. The molecular mechanisms for these suppressive activities is not shown.

Our observations are the first to reveal a role for POLQ in G2/M gap filling – and to show that this occurs through counteracting RAD52. We find RAD52 is required for MRE11 recruitment (in POLQ-depleted cells). Given the extent of the observations (see reviewers comments below) we feel the molecular details, such as POLQ polymerase Vs helicase activities and the interactions of RAD52 itself, for example, are better suited to, likely two separate, follow-up reports.

Overall, I believe that are trying to do too many things in one paper in order to prove two distinct mechanisms. Accordingly, the major impact of the paper is completely lost, and there are now 86 references. The original version of this paper was dense. Now it is even more dense.

The scale of the paper is the result of answering multiple questions from reviewers, including this reviewer. We take the point that having done that, we arguably have two papers, however as the findings are highly related, the clarity of the POLQⁱ differences in BRCA-deficient cells requires the knowledge of the first mechanism of RAD52-suppression in POLQ-depleted cells, the findings exist together. We have separated paragraphs into clear topics and rewritten the abstract and discussion to help the reader.

I have read the abstract of the paper many, many times, on separate occasions, and I cannot make any sense out of it. For instance, what does this sentence from the abstract really mean? “Polθ suppresses RAD52-mediated inhibition of G2/M gap-filling in BRCA1-deficient and 53BP1-deficient cells and RPA:RAD52 encourages MRE11 nuclease accumulation, suppressing DNA synthesis” This sentence is very difficult to interpret.

Our apologies. The sentence is gone and replaced by two, clearer statements. We have rewritten the manuscript with this reviewer (and the reader) in mind.

And at one point in the paper, the authors use five different inhibitors (Olaparib, mirin, ART558, 6-OHD, and USP48 siRNA), to prove these two distinct mechanisms. Genetic arguments would be stronger than drug-related arguments.

These approaches are at the request of other reviewers and exist to complement each other. We have used RAD52 depletion and two inhibitors, we have used mirin and RPA siRNA to assess resection components, as well as mutants of RAD52. The USP48 siRNA is to provide an alternative means to extend resection. For several of the interventions, genetic mutants would fail entirely to progress the findings (RAD52, RPA, MRE11 etc) as the full knockout is either lethal to cells survival or synthetic lethal with other genetics we are investigating. A key conclusion of our findings is the non-digital relationships between RPA-RAD52-MRE11 and POLQ loss or inhibition (expressed well by Reviewer #1 above).

In my opinion, the best way for the authors to prove their two distinct mechanisms is to write two separate papers – one describing mechanism 1, and one describing mechanism 2. Otherwise, a side-by-side comparison of these two mechanisms is not instructive, since the mechanisms are so different.

We agree that it might be possible to write two papers from the considerable findings in this manuscript and the mechanisms are different. However, that difference is only visible by a side-by-side comparison and therefore, we feel the findings belong together. We have rewritten the manuscript with this reviewer (and the reader) in mind.

I wish I could be more positive.

We have rewritten and redrawn critical elements of the paper to improve the readability/understandability of the work.

Reviewer #4 (Remarks to the Author):

I have read the paper a few times now (as it is rather complicated!) and I am convinced with the way the authors deplete and inactivate RAD52.

The fact that both low concentrations and the siRNA and the catalytic inhibitor show exactly the same results in viability as well as different assays (RAD51 foci, breaks, gaps etc) used by the authors convinces me that the results are trustable and valid.

Reading through the manuscript and the answers to the reviewers I am aligning with their interpretations and the conclusions.

Many thanks for the comments – we have rewritten and redrawn critical elements of the paper to improve the readability/understandability of the work.